# Genetically-stable engineered optogenetic gene switches modulate spatial cell morphogenesis in two- and three-dimensional tissue cultures

Hannes M. Beyer [1,6], Sant Kumar [2,6], Marius Nieke [1], Carroll M. C. Diehl [1], Kun Tang[1], Sara Shumka[1], Cha San Koh[1], Christian Fleck[3], Jamie A. Davies [4], Mustafa Khammash [2] ✉ & Matias D. Zurbriggen [1,5] ✉

Recent advances in tissue engineering have been remarkable, yet the precise control of cellular behavior in 2D and 3D cultures remains challenging. One approach to address this limitation is to genomically engineer optogenetic control of cellular processes into tissues using gene switches that can operate with only a few genomic copies. Here, we implement blue and red light-responsive gene switches to engineer genomically stable two- and three-dimensional mammalian tissue models. Notably, we achieve precise control of cell death and morphogen-directed patterning in 2D and 3D tissues by opto-genetically regulating cell necroptosis and synthetic WNT3A signaling at high spatiotemporal resolution. This is accomplished using custom-built patterned LED systems, including digital mirrors and photomasks, as well as laser techniques. These advancements demonstrate the capability of precise spatio-temporal modulation in tissue engineering and open up new avenues for developing programmable 3D tissue and organ models, with significant implications for biomedical research and therapeutic applications.

The field of tissue engineering constantly seeks innovation to recapitalize processes involved in the formation of in vivo tissues in in vitro environments. A plethora of methods has greatly advanced the field, which includes various tissue maintenance protocols, the development of extracellular matrices, and methods for the fabrication of tissues such as bioprinting. However, the external regulation of biological processes with spatiotemporal precision and patterned structures remains a persistent challenge[1].

Biological engineering efforts have greatly advanced our attempts to understand and design biology. Natural and engineered molecular switches enable the regulation of the physiological behavior of biological systems and constitute indispensable routine techniques in industry and fundamental life sciences alike[2,3]. Concepts inherent to synthetic biology brought about design strategies that rendered biology rational and predictive, thereby improving the accuracy with which one may control biology[4,5]. Molecular switches have been engineered to selectively respond to external stimuli ranging from small molecules, temperature, or pH, to optical stimulation[6]. The development of light-responsive molecular tools utilizing a wide variety of natural and engineered photoreceptors to precisely regulate

[1]Institute of Synthetic Biology, Heinrich-Heine-University Düsseldorf, Universitätsstrasse 1, Düsseldorf, Germany. [2]Department of Biosystems Science and Engineering (D-BSSE), ETH Zürich, Klingelbergstrasse 48, Basel, Switzerland. [3]Freiburg Center for Data Analysis and Modeling (FDM), University of Freiburg, Ernst-Zermelo-Straße 1, Freiburg im Breisgau, Germany. [4]Deanery of Biomedical Sciences, University of Edinburgh, Edinburgh, UK. [5]CEPLAS – Cluster of Excellence on Plant Sciences, Düsseldorf, Universitätsstrasse 1, Düsseldorf, Germany. [6]These authors contributed equally: Hannes M. Beyer, Sant Kumar. ✉e-mail: mustafa.khammash@bsse.ethz.ch; matias.zurbriggen@uni-duesseldorf.de

diverse biological activities in orthogonal environments ushered in the field of "non-neuronal optogenetics". In contrast to chemical inducers, light provides high spatial and temporal resolution, reversibility, and facilitates computer control of dynamic biological responses, *e.g.*, by automated programming of light intensity profiles or patterned and pulsatile illumination regimes[7]. To date, optogenetics has found broad application in pro- and eukaryotic organisms, likewise in vitro and in vivo. Examples include the modulation of cell adhesion[8,9], viral infectivity[10,11], extracellular cell signaling[12–14], sub-cellular protein localization and activity[15–20], and the regulation of endogenous and synthetic promoters as well as gene editing[21–30]. Despite the ongoing development and increasing potential of three-dimensional tissue cultures in regenerative and personalized medicine, as well as their growing relevance as alternatives to animal testing, the widespread utilization of optogenetic regulation of biological processes within these cultures has remained largely untapped[31]. For example, the ever-increasing complexity and tissue resemblance of organoid models would greatly benefit from optogenetic modulation because compared to in vivo models, they are much more compatible with optogenetic intervention due to the controlled cultivation conditions and better light permeability of the specimen, as demonstrated recently[32]. As illustrated in this study, creating uniformly light-responsive engineered tissues for both 2D and 3D models requires the implementation of suitable genome engineering strategies.

Optogenetic gene expression systems represent one of the most versatile control strategies known. Rather than individually engineering and optimizing distinct biological modules for optogenetic control, light-regulated promoters can direct the dynamic production of arbitrary bioactive molecules within cells. Despite the time delay resulting from RNA transcription and protein translation, gene switches often represent the easiest way of introducing optogenetic control into a given experimental setup. The use of photoreceptors responsive to distinct portions of the light spectrum also enabled optogenetic transcriptional control of multiple and individual genes, thus providing means to induce various cellular events in selected subsets of cells and reconstruct intertwined functional networks[22,33,34].

Optogenetic gene expression systems are imminently suitable for use in single cells or two-dimensional animal cell cultures, with some striking examples of technology transfer into whole animals[16,24,29,35,36]. Nevertheless, the majority of these applications depend on transiently introducing the required genes into a population of cultured cells or tissues through transfection or transduction. While this strategy suffices for proof-of-principle experiments, the associated mosaicity caused by a heterogeneous distribution of components and temporal system instability due to dilution and degradation effects, discourage the use of transient gene transfer when deriving sophisticated experimental models. For instance, light-guided cell patterning, control of cell viability and morphology, or signal propagation in 3D culture models all necessitate stable genomic integration of the optogenetic tools. Non-transfected, hence non-responsive cells interfere with the integrity of the systems and the spatial resolution.

In this work, we developed an experimental strategy that integrates optogenetic tools, genomic engineering approaches, and spatially precise illumination technologies to derive 2D and 3D mammalian tissue cultures. For this, we first engineered and systematically characterized blue and red light-responsive optogenetic gene expression systems, generating genomically stable mammalian cell lines of various origins. Subsequently, we initiated cellular necroptosis in both 2D and 3D cultures using precise illumination techniques at scales down to micrometers. These techniques involved the utilization of photomasks, laser stimulation, as well as pattern-projection methods such as light sheets and digital-micromirror devices (DMD). Furthermore, we enabled spatial control of cell-cell communication via optogenetic control of WNT3A signaling, crucial for stem cell differentiation and tissue engineering. The resulting cell lines regulated morphogen signaling among three-dimensional spheroid cultures, acting as synthetic WNT3A organizer centers. The tools developed here will serve the future study of WNT3A signaling and necroptosis through optogenetic regulation of gene expression. Our findings offer insights into selecting appropriate optogenetic gene expression systems and their integration as essential elements within mammalian designer genomes. Additionally, they pave the way for largely untapped opportunities in optogenetic stem cell engineering and represent initial steps toward the eventual establishment of optogenetically programmable 3D tissue and organ models.

## Results

### Design of optogenetic gene switches

To develop mammalian cell lines genomically equipped with optogenetic gene expression switches for controlling diverse cell behaviors, we first identified suitable gene switches that remain functional when expressed from a few genomic copies. Strategies to regulate gene expression in mammalian cell systems optogenetically have been devised using a variety of different photoreceptors with unique responses to light of different colors. Systems sensitive to UVB[33], blue[17,22,24,33,37–42], green[25,34], and red/far-red[21,28,34,43,44] light have allowed individual genes to be controlled using four main architectural principles: protein dimerization[21,22,28,34,37,41], DNA binding[24,25], protein localization[16,17,42], and protein clustering[25,39,43]. We focused on variants of red/far-red and blue light-inducible gene expression systems that rely on the induction of protein heterodimerization. We concentrated here on systems we have developed or tested over the past years using transient transfection approaches and provide us with an easy-to-estimate quantitative output. In particular, we selected those for which we would expect either good dynamic induction ranges[22], reversibility[21], or low basal expression levels based on our experience (Fig. 1A, B). Additionally, we included a single-component blue light system relying on light-regulated modulation of DNA binding, because it has a simpler architecture and fewer required components (Fig. 1C).

We used the Sleeping Beauty 100X transposase for genomic integration of the required components into common mammalian cell lines, namely CHO-K1, HEK-293, HEK-293T, and HeLa[45,46]. Similar to lentiviral gene-integration methods but without size limitation, transposases may accomplish multiple genomic insertion events of the introduced transgene at random positions, thus increasing the chance of identifying well-performing clones with sufficient expression strength. Furthermore, transposase systems do not necessitate the production of viral particles, further simplifying the experimental protocol.

We engineered two variants of red/far-red light-regulated gene switches for which we tested two orthogonal DNA-binding proteins and three variants of blue light switches. We compared gene delivery of the required components either based on their successive genomic transposition or using multicistronic transcripts (Fig. 1A–E). For a detailed explanation of the architecture and mode of function of the gene switches, please refer to Fig. S1 and the Note to Figure S1 in the Supplementary Information. The red/far-red light switches RED$_{TET}$ and RED$_E$ constitute light-regulated split transcription factor systems based on the photoreceptor phytochrome B from *Arabidopsis thaliana* and a minimized binding sequence derived from the phytochrome-interacting-factor 6 (PhyB$_N$/PIF6$_{APB}$) equipped either with TetR or E as DNA-binding protein (Figs. 1A and S1A, Supplementary Information)[21,47]. Similarly, the BLUE$_{SINGLE}$ and BLUE$_{DUAL}$ switches act as split transcription factors utilizing the heterodimerization of an engineered LOV2 domain from phototropin 1 of *Avena sativa* and an ePDZb domain in response to blue light[22,48]. Here, we tested whether genomic transposase-mediated integration of the split components on individual vectors or combined in a single transcript would result in a higher chance of isolating an optimal clone from the resulting

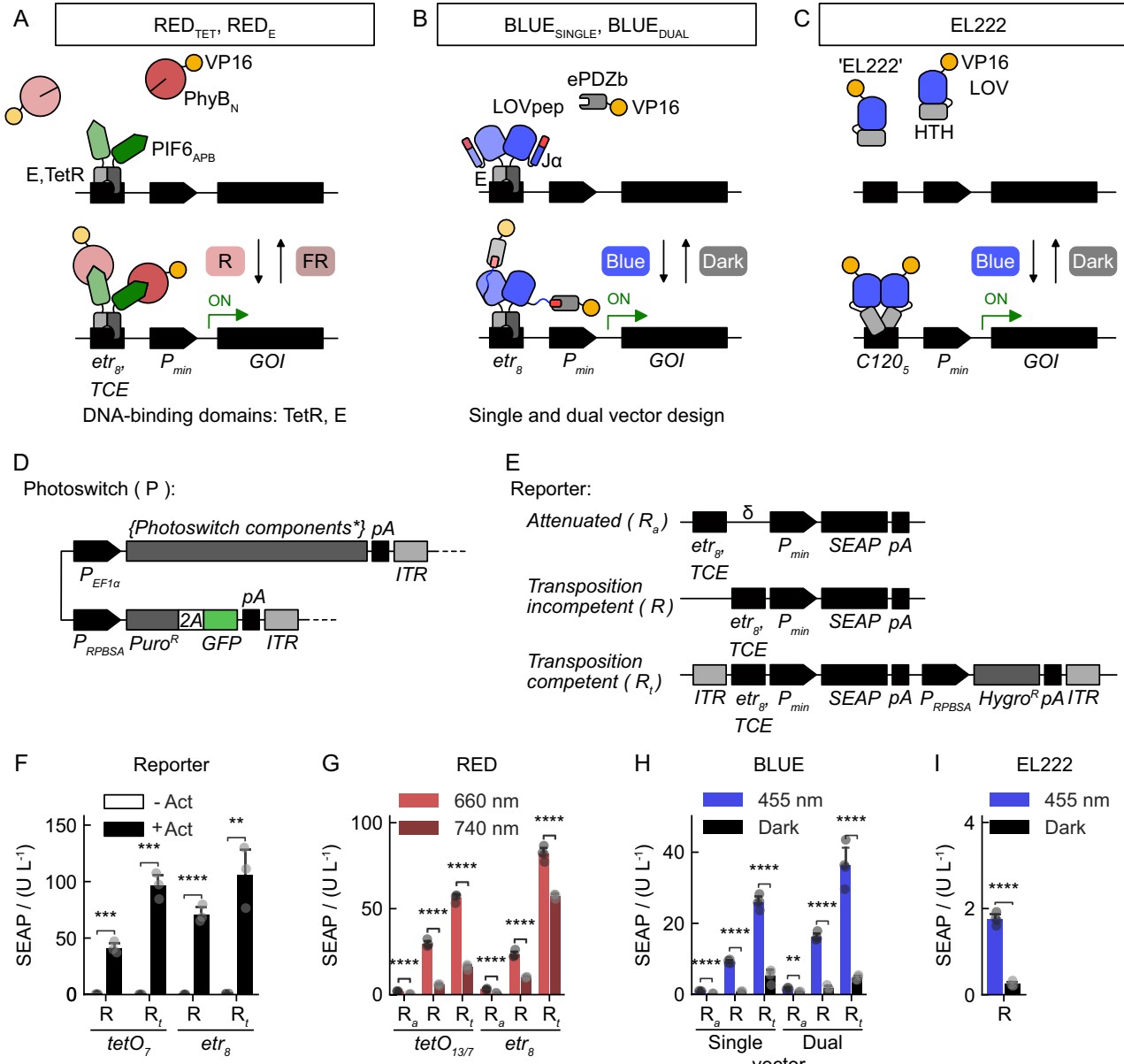

**Fig. 1 | Design of optogenetic switches for genomic cell engineering.**
**A**–**C** Modes of function for the optogenetic gene switches. Light exposure activates the expression of a gene of interest (*GOI*) via recruitment of the transactivation domain VP16 to a minimal CMV promoter (P_min). **A** RED gene switches utilize the PhyB_N/PIF6_APB interaction and either E or TetR as the DNA-binding domain specific for the *etr_8* or *TCE* DNA elements, respectively. Red light (R) activates PhyB_N and initiates binding to PIF6_APB, and thus recruitment of the reconstituted TF to the target sequence of the respective DNA-binding domain to the synthetic promoter, whereas far-red light causes dissociation. **B** BLUE gene switch implemented either on one (SINGLE) or two (DUAL) independent vectors. Blue light causes the binding of LOVpep to ePDZb via relaxation of helix Jα exposing the otherwise caged ePDZb-interacting peptide. **C** The EL222 single-component gene switch exhibits natural photo-regulated DNA binding capacity to the *C120* operator sequence in response to blue light. Fusion of EL222 to VP16 results in a functional optogenetic TF for eukaryotic cells. **D** General architecture of transposition-competent vectors encoding photoswitches. **E** Architecture of attenuated (R_a, δ spacing, no ITRs), transposition-incompetent (R, no ITRs), and transposition-competent (R_t) reporter constructs utilizing *SEAP* as the *GOI* (see Table S1, Supplementary Information, for details). *TCE* contains seven *tetO* repeats. **F** Functional tests of tTA (TetR-VP16) and eTA (E-VP16)-activatable SEAP reporter constructs as in (**E**) using transient transfection together with the respective activators (Act) into CHO-K1 cells. *p*-values: *tetO_7*-R, 1.622e⁻⁴; *tetO_7*-R_t, 1.319e⁻⁴; *etr_8*-R, 8.976e⁻⁵; *etr_8*-R_t, 2.577e⁻³. **G** Transient transfection test of transposition-competent vectors for the RED_TET and RED_E system as in (**D**) for E and TetR-specific systems in CHO-K1 cells. The attenuated TetR-specific reporter construct contained 13 *tetO* operator repeats. *p*-values: *tetO_13*-R_a, 2.332e-7; *tetO_7*-R, 2.331e-7; *tetO_7*-R, 6.052e-8; *etr_8*-R_a, 1.203e-5; *etr_8*-R, 7.520e-6; *etr_8*-R_t, 1.922e-5. **H** Transient test of BLUE_SINGLE and BLUE_DUAL transposition-competent vectors in CHO-K1 cells as in (**D**). *p*-values: Single-R_a, 1.514e-6; Single-R, 6.492e-8; Single-R_t, 4.344e-6; Dual-R_a, 1.027e-3; Dual-R, 6.120e-7; Dual-R_t, 3.229e-5. **I** Transient test of the transposition-competent vector encoding the EL222 photoswitch in CHO-K1 cells. *p*-value: 8.224e-7. **F**–**I** Used reporter constructs as in (**E**). Data represent mean values with one standard deviation of four biological replicates compared with two-sided independent Student's *t*-tests. **, 1e-3 <*p* ≤ 1e-2; ***, 1e-4 <*p* ≤ 1e-3; ****, *p* ≤ 1e-4. Source data are provided with this paper.

polyclonal culture (Figs. 1B and S1B, Supplementary Information). Lastly, we tested another blue light system derived from EL222 from *Erythrobacter litoralis*, which acts as a single molecular component where photoexcitation directly regulates the DNA-binding affinity to a specific sequence termed *C120* (Figs. 1C and S1C, Supplementary Information)[24].

To integrate the required transgenes, we generated vectors capable of undergoing genomic transposition: these vectors encoded the

genes required for the photoswitches controlled by a constitutive $P_{EF1\alpha}$ promotor (Figs. 1D and S1, Supplementary Information)[45]. In addition, we designed target vectors encoding human placental secreted alkaline phosphatase (SEAP) as a quantitative reporter under the control of inducible promoters matching the DNA-binding properties of the photoswitches, namely specific for E, TetR, or EL222. For the EL222-based system, we exclusively tested transposition-incompetent reporter vectors (Fig. 1E, label 'R'). These target vectors contained the TCE (tetracycline response element and minimal CMV-promoter enhanced) promoter comprising seven tetracycline (tet) operator repeats[45], or a hybrid promoter consisting of eight erythromycin repressor operator sites placed upstream of the minimal cytomegalovirus promoter ($P_{min}$) (Fig. S1, Supplementary Information). Transient transfection experiments using CHO-K1 cells confirmed that the constructs were able to raise an optogenetic response (Fig. 1F–I). Tested with the constitutive transcriptional activators tTA (TetR-VP16) and eTA (E-VP16), transposition-competent vectors ($R_t$) produced slightly stronger responses compared to previously described reporters (R) (p-value: $tetO_7$, 0.0016; $etr_8$, 0.099, two-sided independent Student's t-tests)[21,33,49], probably caused by the presence of an additional downstream promoter element required for selection markers. None of the reporter constructs showed signs of relevant basal expression in the absence of a transactivator. Optogenetic experiments with red/far-red (660/740 nm, Fig. 1G) and blue (455 nm, Fig. 1H, I) light using a transient transfection protocol confirmed that the vector combinations resulted in functional light-responsive gene switches. Again, transposition-competent vectors responded stronger, overall, than incompetent ones and showed an increased background activity for the RED systems (Fig. 1G, H). However, we speculated that a drastic reduction in the copy number upon genomic transposition of the constructs would probably compensate for this effect and reduce the basal activation in the non-illuminated state.

## Partial genomic transposition of optogenetic gene systems

Next, we stably integrated $P_{EF1\alpha}$-driven constitutive expression cassettes coding for the optoswitches through genetic transposition and puromycin and blasticidin selection into the genomes of CHO-K1, HEK-293, HEK-293T, and HeLa cells (Fig. 2A–E and Fig. S2, Supplementary Information). The vectors encoded six variants of optogenetic transcription factors, resulting in 20 polyclonal cell cultures, each representing pools of heterogeneous cells possessing unique genomic insertion events. We tested the cell cultures for their averaged responsiveness in inducing matching synthetic promoters upon light stimulation first using a transient transfection approach for the

reporter plasmids (Figs. 2A–E and S2, Supplementary Information). We speculated that the presence of the constitutive RPBSA promoter, transcribing the genes required for antibiotic selection, could partially contribute to the higher background promoter activity in transposition-competent vectors (see additional notes to Fig. S2, Supplementary Information) and tested different vector amounts in the transfection tests.

Among the four hosting cell lines, CHO-K1 showed the overall best optogenetic induction of the SEAP reporter production with the lowest background levels upon light stimulation. HEK-293T cells showed the strongest SEAP expression levels overall (Fig. S2, Supplementary Information). HeLa cells responded poorly to light induction for all tested optogenetic systems, though a minor expression induction was noticeable in the cells that encoded BLUE systems. All reporter constructs, however, could be activated by co-transfecting a light-independent transcriptional activator that utilized the VP16 domain. HEK-293 cells either displayed relatively weak expression levels in the illuminated state with SEAP levels below 20 U/L ($RED_{TET}$) or the highest background reporter production among the tested conditions, ranging around 100 U/L of SEAP production giving rise to red light-specific induction levels below a factor of two ($RED_E$). The results presented in Figs. 2 and S2, Supplementary Information, are strongly influenced by phenomena resulting from transfection efficiency differences between the cell lines and the above-mentioned imbalance in genetic copy numbers. Therefore, this data set is no direct measure of the performance of the individual switches upon full genomic integration and clonal selection with respect to induction and expression strength. However, the data provide estimates and also allow a certain degree of ranking (Fig. S2, Supplementary Information). Despite intrinsic beneficial architectural properties, EL222 consistently gave the lowest GOI expression levels (below 4 U/L SEAP production upon blue light exposure), but induction levels in the range of 10-fold could be observed. BLUE systems performed best overall and showed activity in most cell lines (except for HeLa). Red light systems were only marginally active in HEK-293T cells.

## Complete genomic transposition of optogenetic gene systems

In order to generate fully genomically self-contained optogenetic cell lines, we chose to integrate the E-specific SEAP reporter plasmid into the two CHO-K1 cell lines harboring either $BLUE_{SINGLE}$, or $BLUE_{DUAL}$ (Figs. 3A, 2C, D for reference). Both resulting polyclonal cultures showed prominent induction characteristics after illumination with blue light (Fig. 3A, grey-shaded bars) with an 84-fold culture-averaged enhanced SEAP concentration for $BLUE_{SINGLE}$. To evaluate whether the

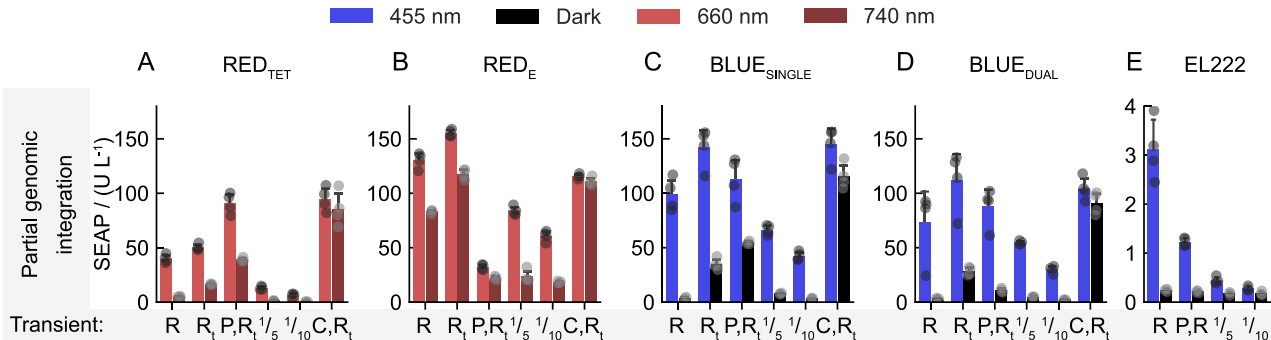

**Fig. 2 | Partial genomic integration of optogenetic gene switches. A–E** Cultures represent a mixed, polyclonal population of cells, selected with antibiotics, with individual genomic insertion events of the photoswitch constructs as in Fig. S1, Supplementary Information. Additionally, the cultures were transiently transfected with the reporter constructs R or $R_t$ (see Fig. 1E), the photoswitch-encoding vector P which was used for the generation of the culture, a 1/10 or 1/5 reduced amount of $R_t$, or C, the constitutive activators tTA or eTA in combination with $R_t$. Cultures were illuminated for 24 h with 455 nm light at an intensity of 10 μmol m⁻² s⁻¹, or with 660 or 740 nm light at 20 μmol m⁻² s⁻¹, prior to determining the secreted SEAP levels. CHO-K1-derived cultures are shown, see Fig. S2, Supplementary Information, for the full dataset with all cell lines. Data represent mean values with one standard deviation of four biological replicates (three replicates for the 455 nm light sample of $BLUE_{SINGLE}$ with 1/10 dilution). Source data are provided with this paper.

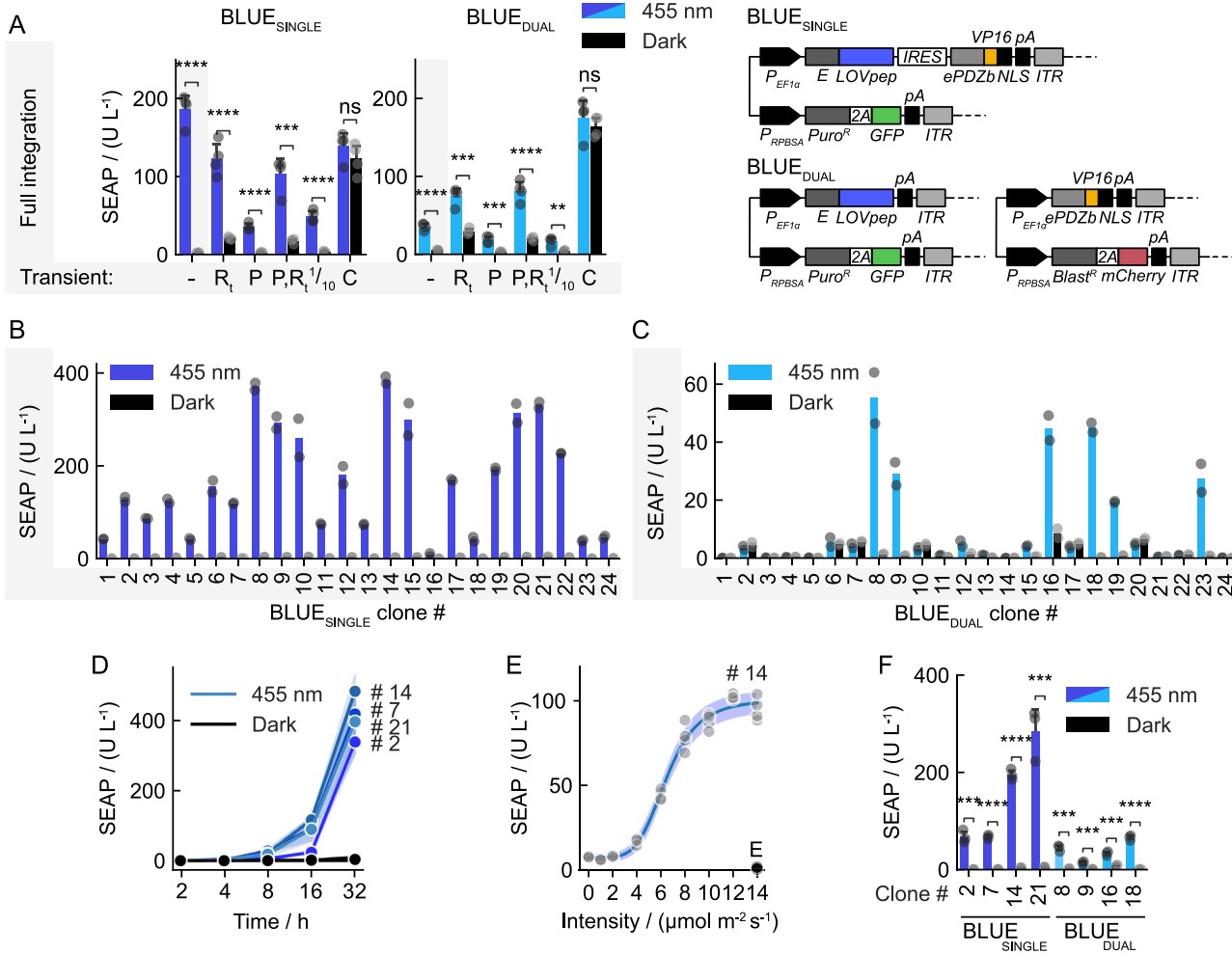

**Fig. 3 | Complete genomic transposition of optogenetic gene systems. A** CHO-K1-derived bulk cultures after complete genomic transposition of the BLUE$_{SINGLE}$ (dark blue) or BLUE$_{DUAL}$ (light blue) vectors and matching R$_t$ SEAP reporter constructs (see Fig. 1E). Gray shaded bars highlight fully genomically-stable conditions without any additional plasmid transfection. The cultures were illuminated for 24 h with 10 μmol m$^{-2}$ s$^{-1}$ light of 455 nm or kept in darkness. Cultures were left untreated or additionally transfected with the indicated components as in Fig. 2 prior to optogenetic tests. The used vector architectures are shown to the right. *p*-values (BLUE$_{SINGLE}$): -, 1.479e-6; R$_t$, 8.396e-5; P, 4.997e-6; (P,R$_t$), 3.000e-4; $^1/_{10}$ R$_t$, 1.620e-5; C, 2.645e-1. *p*-values (BLUE$_{DUAL}$): -, 9.682e-6; R$_t$, 2.547e-4; P, 4.498e-4; (P,R$_t$), 8.867e-5; $^1/_{10}$ R$_t$, 2.614e-3; C, 4.377e-1. **B** Profile of randomly selected clones derived from the BLUE$_{SINGLE}$ vector culture. Data points represent SEAP values determined from a single sample for each clone measured with two different dilutions of culture medium (technical replicates), the bar indicates the average. **C** Experiment as in (**B**) using randomly selected clones derived from the BLUE$_{DUAL}$ vector culture.

**D** Kinetic response profile of the indicated clones in (**B**) using 455 nm light at an intensity of 10 μmol m$^{-2}$ s$^{-1}$ over 32 h. Mean values of four biological culture replicates are shown with error bands representing one standard deviation. **E** Dose response curve using clone #14 with blue light intensities in the range of 0 – 14 μmol m$^{-2}$ s$^{-1}$. Data obtained from four biological replicates were fitted into a Hill binding isotherm (blue line) yielding an $I^{50}$ intensity value of 6.45 μmol m$^{-2}$ s$^{-1}$. Data points marked with "E" at the highest intensity were obtained from samples supplemented with 2 μg/mL erythromycin. Unaveraged single data points are plotted, the error band represents the 95% confidence interval of the fit. **F** Comparison of selected clones of (**B**) and (**D**). *p*-values: #2, 8.419e-4; #7, 7.405e-6; #14, 6.692e-6; #21, 8.748e-4; #8, 3.166e-4; #9, 6.138e-4; #16, 5.813e-4; #18, 2.548e-5. **A, F** Data represent mean values with one standard deviation of four biological replicates compared with two-sided independent Student's *t*-tests. ns, 5e-2 <*p* ≤ 1; **, 1e-3 <*p* ≤ 1e-2; ***, 1e-4 <*p* ≤ 1e-3; ****, *p* ≤ 1e-4. **B, C** Mean values of two technical replicates are shown. **A–F** Source data are provided with this paper.

best performance was achieved, we additionally transiently transfected the plasmids encoding either the photoswitch or the SEAP reporter, however, this weakened the total expression strength or the induction ratio (or both). When comparing the averaged cell cultures, BLUE$_{SINGLE}$ performed much better than BLUE$_{DUAL}$, with a 5.3-fold stronger SEAP production upon light induction (187 ± 20 vs. 35 ± 4 U/L). This induced expression strength reached with BLUE$_{SINGLE}$ was comparable to the levels obtained upon transient transfection of a constitutive activator (C), which was not achieved with BLUE$_{DUAL}$, where the levels remained five-fold lower. Both cultures could still be activated in a light-independent manner to comparable levels by transiently transfecting a plasmid encoding the constitutive E-VP16 transactivator, indicating a comparable representation of the reporter construct within the genomic space of the two cell cultures. Of note, basal expression levels

in uninduced conditions were much lower than in transient experiments (Fig. 2), presumably due to the lower copy number and the construct linearization upon genomic transposition.

We then investigated the cause of the different performance levels of BLUE$_{SINGLE}$ and BLUE$_{DUAL}$ after full genomic integration of the components. We hypothesized that the probability of inserting both constructs of the BLUE optogenetic transcription factor, E-LOVpep, and ePDZb-VP16-NLS, both at genomic loci within the same cell conferring reasonably high expression levels, is probably relatively low. This means that on average BLUE$_{DUAL}$ cells in the bulk culture would respond much weaker compared with BLUE$_{SINGLE}$. Due to the larger combinatorial space resulting in cells with variations of expression levels for the two required components, the chance might exist to isolate a clone from the BLUE$_{DUAL}$ bulk culture with better

performance than any of the individual cells present in the BLUE$_{SINGLE}$ culture. Additionally, BLUE$_{DUAL}$ might enhance the chances of isolating clones with desired characteristics such as basal expression levels, or induction ranges. We randomly isolated 24 clones from each of the two polyclonal cultures (Fig. 3B–F). All BLUE$_{SINGLE}$ clones showed a clear induction upon blue light treatment, albeit at different strengths spanning a 34-fold difference between the weakest (#16) and strongest clone (#14) (Fig. 3B). The clones isolated from the BLUE$_{DUAL}$ culture, however, largely remained non-responsive to blue light or responded relatively weakly (Fig. 3C). Six clones out of 24 responded with weak to medium induction levels to blue light compared with the clones isolated from the BLUE$_{SINGLE}$ culture, with the strongest clone (#8) showing about 16-fold lower expression strength than the average of the BLUE$_{SINGLE}$ clones. The kinetic response profile of four BLUE$_{SINGLE}$ clones at medium and strong induction levels behaved similarly (Fig. 3D). A dose-response experiment with BLUE$_{SINGLE}$ clone #14 treated with blue light intensities ranging from 0 – 14 $\mu$mol m$^{-2}$ s$^{-1}$, revealed an $I^0$ intensity of 6.45 $\mu$mol m$^{-2}$ s$^{-1}$ (Fig. 3E). A direct comparison between selected BLUE$_{SINGLE}$ and BLUE$_{DUAL}$ clones further validated the trend (Fig. 3F). The data suggest that the inclusion of all genetic components involved in single vectors, instead of being split between vectors, is a beneficial strategy for a successful genomic transposition into appropriate loci, which gives rise to the best system performance. Of note, the strategy simplifies the experimental protocol by avoiding an additional antibiotic selection and served as the experimental basis for 2D and 3D control of mammalian cell fates and patterned tissue morphogenesis experiments in this work, as described below.

## Optogenetic induction of necroptosis in 2D and 3D mammalian tissue cultures

Cell differentiation and proliferation represent essential processes during animal development. Next to cell division control, the fine spatiotemporal regulation of various forms of programmed cell death underlies the formation and homeostasis of tissues at every developmental stage. Two pathways of regulated cell death in animals, necroptosis and pyroptosis, involve lytic mechanisms causing inflammatory responses, whereas, in the case of apoptosis, the cells retain the integrity of the plasma membrane and are removed by phagocytes without inducing inflammation. Given the pivotal and general role in tissue shape definition and cell homeostasis through cell removal, the molecular basis of the various forms of cell death and their regulation within organisms is an active field of research[50]. To enable a precise study of signaling pathways and for their use in tissue engineering efforts, optogenetic control over pathway induction represents a great asset to the field, as it opens the possibility of inducing cell death with spatiotemporal precision similar to that seen during developmental tissue formation. A common principle of the natural stimulation of the pathways relies on protein dimerization or oligomerization[51–54]. Caspases, for example, deeply engage in the onset of cell apoptosis and pyroptosis. Initiator caspases activate from a zymogenic form through proteolytic self-cleavage upon dimerization. Oligomerization of the pseudokinase MLKL (mixed lineage kinase domain-like) causes necroptosis through plasma membrane rupture. Consequently, optogenetic approaches that mimic these activation mechanisms have recently been developed, for example, utilizing the induced oligomerization of the blue light photoreceptor cryptochrome 2 from *Arabidopsis thaliana* or a circularly permuted *Avena sativa* LOV2 domain which inhibits the oligomerization in the dark state[55–57].

Controlling the expression of a constitutively active effector of cell death represents a straightforward alternative approach that avoids tedious optimization of the death effectors and the photoreceptors, or dealing with high basal cytotoxic activities of the effective components in the absence of light intrinsic to the strategies involving their direct photoactivation[56]. Utilizing such tools in mid- to

long-term experiments or even for application in three-dimensional tissues, requires stable genomic integration of the components into a host cell, which would otherwise strongly suffer from interference of wild-type cells that remain unresponsive despite photoactivation. For prolonged cultivation prior to the optical control, a protective mechanism that renders the specimen insensitive to light should therefore exist.

We tested whether we could derive an engineered cell line using some of the photoswitches in Fig. 2 and Fig. S2, Supplementary Information, for optogenetically inducing necroptotic cell death. For this, we chose the BLUE$_{SINGLE}$ and RED$_E$ vector systems. The E protein as the DNA-binding domain in these constructs allows erythromycin addition for safe cultivation and serves as a control to distinguish optogenetically-induced programmed cell death from simple phototoxicity. Erythromycin inhibits DNA binding of the E protein transcription factor through allosteric regulation[47].

It has been shown that the expression of the N-terminal four-helix bundle domain of MLKL suffices to induce necroptosis, causing oligomerization and plasma membrane rupture along with exposure of phosphatidylserine on the cell surface and induction of calcium influx[56]. Therefore, we replaced the SEAP gene from the engineered transposon vector used in Fig. 2 with a sequence encoding the N-terminal 178 residues of MLKL (MLKL$_N$, Fig. 4A). We derived stable polyclonal cultures with the Sleeping Beauty 100X transposase and puromycin/hygromycin selection using CHO-K1 cells. Initial tests indicated that the cultures indeed contained a fraction of responsive cells undergoing cell death upon illumination with 10 $\mu$mol m$^{-2}$ s$^{-1}$ blue light, or 20 $\mu$mol m$^{-2}$ s$^{-1}$ red light for 24 h (LED light overall illumination) (Fig. S3, Supplementary Information). We then isolated functional clones for further microscopic studies (clone #10 for BLUE$_{SINGLE}$, referred to as CHO-K1$^{NecrOpto}$). When illuminated with either blue (455 nm) or red (660 nm) light, the respective cell lines consistently died after 24 h of illumination (Fig. 4B, blue; Fig. S4, Supplementary Information, red). Importantly, the addition of erythromycin provided reliable protection with comparable cell survival to that observed for samples incubated in darkness. This indicated that the used light intensities of 10–20 $\mu$mol m$^{-2}$ s$^{-1}$ did not cause phototoxic effects. This further confirmed that the induced cell death in illuminated samples without erythromycin supplementation indeed resulted from the molecular action of the produced MLKL$_N$. A dose-response curve determined an $I^0$ blue light intensity value with half-minimal cell survival of 3.03 $\mu$mol m$^{-2}$ s$^{-1}$ (Fig. 4C).

Two-dimensional adherent tissue cultures are intrinsically well suited for spatiotemporal optogenetic experiments because of (i) their restricted cellular mobility, (ii) the thin tissue layer allowing homogeneous exposure to a light source, and (iii) compatibility with live-cell imaging in a single focal plane. However, we questioned whether we could also induce necroptosis in three-dimensional tissues, very relevant to applications related to tissue engineering. We prepared three-dimensional spheroid cultures using CHO-K1$^{NecrOpto}$ cells and exposed them to all-over illumination with 455 nm blue light followed by microscopic endpoint assessment after one day (Fig. 4D). Illuminated spheroids showed severe morphological changes across z-stacks, where the outer layer of cells started dissociating from the 3D tissues. The observation was accompanied by an overall loss of EGFP fluorescence and a strong gain in blue fluorescence arising from SYTOX Blue, which we used to stain the nuclei of dead cells. Erythromycin-supplemented cultures, however, remained viable, again very similar to samples kept in the dark.

Next, we questioned whether we could precisely define alive and death zones within tissue cultures on command with spatially restricted illumination. We first resorted to a photomasking strategy in a two-dimensional CHO-K1$^{NecrOpto}$ culture whereby we projected blue light from the bottom through a 3D-printed mask positioned underneath each well of a 24-well plate. We used the previously developed light

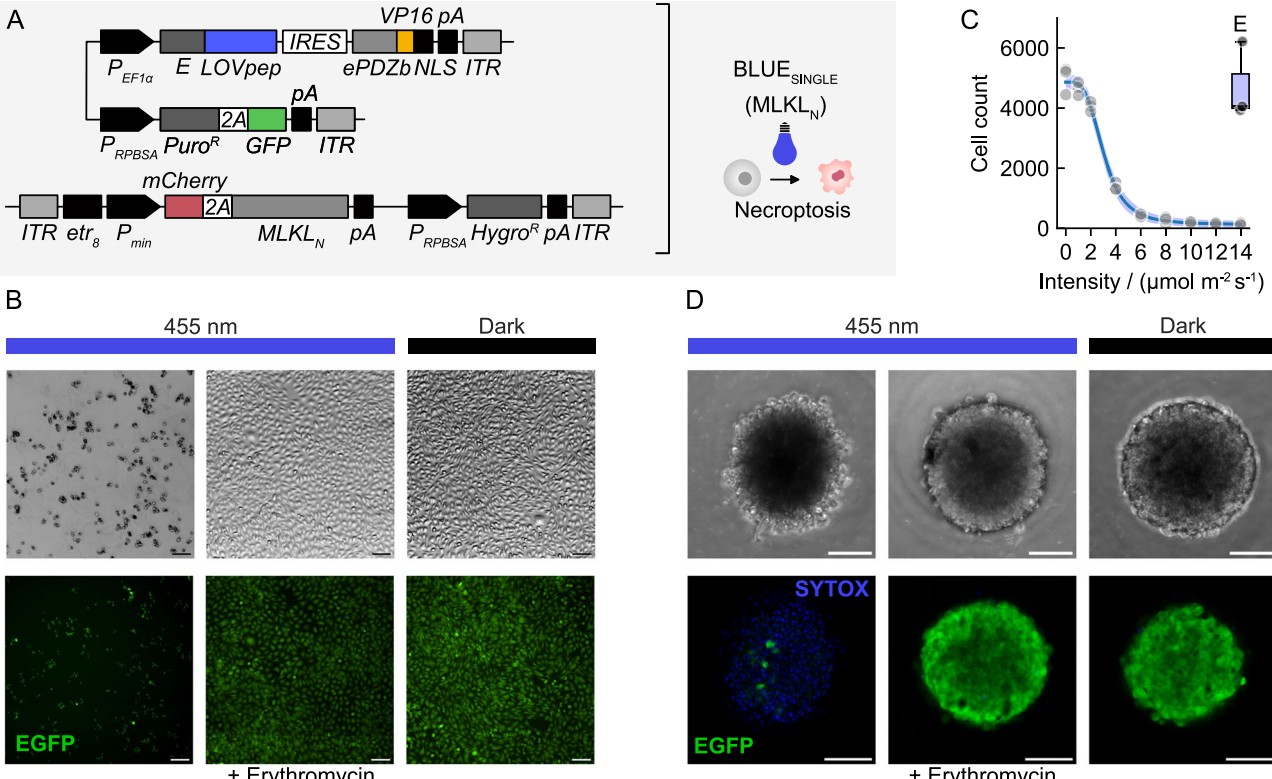

**Fig. 4 | Optogenetic induction of necroptosis in 2D and 3D mammalian tissue cultures. A** Genetic constructs enabling blue light-induced expression of *MLKL_N* for the subsequent induction of necroptosis based on BLUE$_{SINGLE}$ and the vector design in Fig. 1D, E. See Fig. S4, Supplementary Information, for the RED$_E$ system. Inverted brightfield images are shown. **B** Induction of necroptosis after 24 h of overall LED blue light illumination with an intensity of 10 μmol m$^{-2}$ s$^{-1}$ using a CHO-K1-derived clone (clone #10) in a 2D cell culture. The addition of 2 μg/mL erythromycin confers protection from necroptosis. The experiment was performed seven times with similar results. **C** Dose-response curve of optogenetic necroptosis induction using clone #10. The living cell number in a microscopic area of 1.76 mm$^2$ was determined after 24 h of blue light illumination using the indicated intensities. Data obtained from three biological culture replicates were fitted into a Hill binding isotherm (blue line) yielding an $I^{50}$ intensity value of 3.03 μmol m$^{-2}$ s$^{-1}$. Unaveraged single data points are plotted, the error band represents the 95% confidence interval of the fit. Data points labeled with "E" at the highest intensity derive from samples supplemented with 2 μg/mL erythromycin, overlaid with a box plot showing the 25$^{th}$, 50$^{th}$, and 75$^{th}$ percentiles where the whiskers extend to points that lie within a 1.5 interquartile range. Source data are provided with this paper. **D** Induction of necroptosis after 24 h of blue light illumination in 3D spheroid cultures using the cells as in B with a blue light intensity of 20 μmol m$^{-2}$ s$^{-1}$. Brightfield and confocal EGFP and SYTOX microscopic images of the same sample are shown. The experiment was performed three times with similar results. **B**, **D** Scale bar, 100 μm; representative images are shown.

plate apparatus (LPA) which greatly facilitated the mounting of the mask due to its sandwich design and the possibility of varying the light intensity in every well (Fig. S5A, Supplementary Information)[58]. A set light intensity of 40 μmol m$^{-2}$ s$^{-1}$ reaching the center of the photomask resulted in the sharpest separation of survival and death zones in a two-dimensional culture (Figs. 5A, S5B, Supplementary Information).

While photomasking does not provide a means to flexibly illuminate distinct regions within cultures, it suffices to demonstrate the ability to define patterns of responsive cells through spatial illumination. To select distinct culture areas for illumination and to gain insights into the kinetics of spatial necroptosis induction through modulation of light intensities combined with live cell imaging, we utilized our custom-designed DMD (Digital Micromirror Device)-based platform – μPatternScope[59]. This platform, when optically coupled with a microscope and a light source, defines high-resolution LED blue light patterns reflected from a DMD chip and focuses them onto the sample plane of the microscope. μPatternScope allows concurrent spatial and time-resolved photostimulation and imaging of cells while modulating a range of light intensities over the target area under the microscope (Fig. 5B, C and Movie S1, 2, Fig. S6, Supplementary Information). The projected pattern consisted of several elements. The top area contained four rectangles of about 400 μm x 600 μm, each. Gray-scaling allowed modulation of the light intensity in each area within the 8-bit depth (255 shades). The bottom area was used to project a 0-255

shade gradient and the center for dynamic modulation of the "hhu" Düsseldorf University logo with each of the three letters appearing in time intervals of 10 h. The LOV2 domain used in the BLUE gene switch has an excited state half-life time of about 17 s after which ePDZb binders dissociate quantitatively within a few minutes in vitro[22,48]. The fast dissociation kinetics of the photoswitch heterodimer and a continuously tunable photoexcitation lifetime enable fine-tuning of the expression strength in the BLUE switch upon adjusting the light intensity. However, productive transcription events probably require a prolonged or continuous excitation and immobilization of the sample when performing spatially defined experiments. These features, intrinsic to the *As*LOV2 domain, therefore render samples relatively insensitive to short-term exposure to light, such as during intervals of live cell imaging, even when using white or blue light for excitation. We chose 1 h image acquisition intervals during constant blue light pattern projection for 33 h to allow thermal relaxation of photoreceptor molecules in dark zones after each acquisition (Fig. 5B, C). After 3 h of pattern projection, we observed initial mCherry signals arising with sharp edges at high projection intensities, and soft edges at low projection intensities. This suggests that gray shade scaling is an appropriate approach for modulating the necroptotic effect throughout the sample field. In the region with the highest projection intensity, we observed indications of progressive necroptosis in the brightfield channel as early as 4 h, preceded by a rise of mCherry signal, before

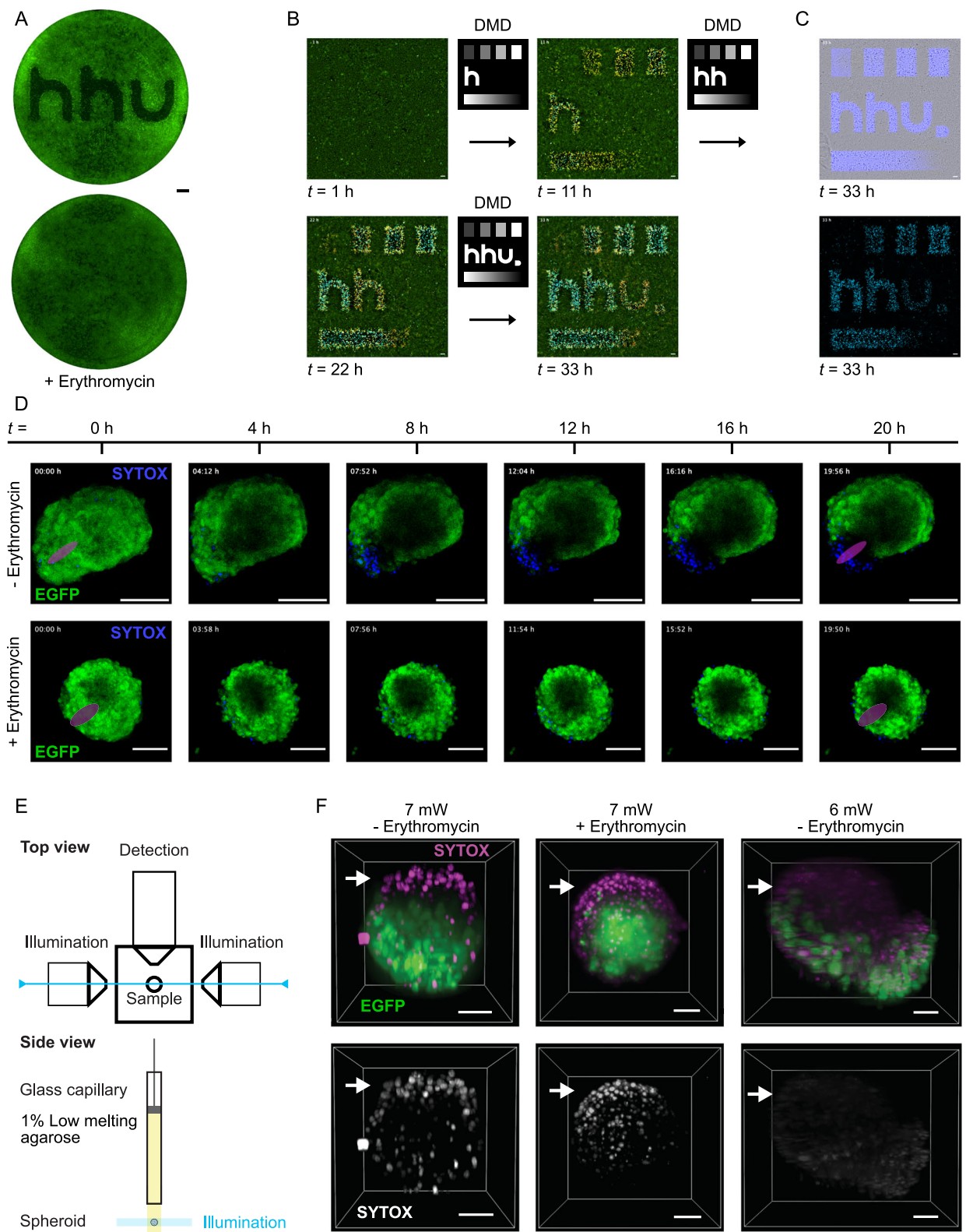

gradually developing into SYTOX Blue positive events with a delay of another 4 h (Movie S1, Supplementary Information). Also in this experiment, the addition of erythromycin provided an effective shielding mechanism from the induction of gene expression and necroptosis (Fig. S6, Movie S2, Supplementary Information).

The spatiotemporal optogenetic induction of cellular functions in a 3D tissue is rather challenging compared with 2D cultures. The experiment poses particular challenges to the microscopic analysis and induction due to the relatively thick and freely floating tissues. However, given the technical challenges are manageable, *e.g.*, through matrix-embedding of 3D cultures, we succeeded in defining death zones within CHO-K1[NecrOpto] spheroid tissues using confocal 488 nm laser excitation, where the scanning frequency supported continuous activation of the BLUE[SINGLE] switch (Fig. 5D, Movie S3 and Fig. S8,

**Fig. 5 | Spatiotemporal control of 2D and 3D tissues. A** Spatial induction of necroptosis using a 3D-printed photomask mounted in the LPA device using a light intensity of 40 μmol m⁻² s⁻¹ (blue LED light). The bottom sample shows the erythromycin control at identical conditions (see Fig. S5, Supplementary Information). Scale bar, 1 mm. The experiment was performed three times with similar results. **B** Spatial, dynamic, and quantitative control of optogenetic necroptosis induction using the digital mirror device for patterning using a 4x magnification objective in a time series. 8-bit grey shades correspond to 60, 120, 180, and 255. An overlay of EGFP, mCherry, and SYTOX Blue signals is shown. See Fig. S6 and Movie S1–2, Supplementary Information, for controls and movies. Scale bar, 100 μm. **C** Brightfield and blue light pattern projection (top) and SYTOX Blue signal (bottom) of the experiment in (**B**) at 33 h. **D** Spatial induction of necroptosis in a 3D culture of CHO-K1^NecrOpto cells using 488 nm laser excitation over a period of 20 h

shown as intensity projection of a Z-stack. The tissue sample in the bottom row was supplemented with 2 μg/mL erythromycin. See Fig. S8, Supplementary Information, for individual z-layers. Scale bar, 100 μm. Each experiment was performed once, see Fig. S7, Supplementary Information, for a repeat with higher imaging frequency. **E** Schematic representation of spheroid blue light illumination and imaging using a custom light sheet microscope. **F** Spatiotemporal regulation attempts of necroptosis in 3D tissues using light sheet microscopy. 3D spheroid cultures derived from CHO-K1^NecrOpto cells were illuminated with a plane (white arrows) of 488 nm laser light for optogenetic activation at the indicated intensities. After 5 h, z-stacks were acquired visualizing constitutive EGFP fluorescence and cell death by staining with SYTOX Red. The experiments were performed four times independently. Scale bar, 50 μm.

Supplementary Information). Also in this experiment, erythromycin addition protected the samples from induced necroptotic effects (Fig. 5D, Movie S4, Supplementary Information). When we tried to record partial confocal z-stacks using the 405 and 488 nm lasers and additionally brightfield stacks combined with EGFP and SYTOX Blue epifluorescence imaging in 30 min intervals, we apparently reached the limitations of light exposure tolerance. Besides the local induction, we observed a significant general increase in necroptotic tissue at the spheroid surface, gradually progressing over time (Fig. S7, Movie S5, Supplementary Information).

Lastly, to enable planar optogenetic modulation of 3D tissues, we tested the photoactivation of necroptotic spheroid tissues using light sheet microscopy. In this approach, we aimed to deliver z-plane restricted spatial blue light activation to 3D CHO-K1^NecrOpto spheroid tissues (Fig. 5E, F) for which we used a custom microscope (a Gaussian light sheet constructed using a cylindrical lens, double-sided excitation lasers, and a single-sided detection orientation) to illuminate one plane of the spheroids with a wavelength of 488 nm. However, across laser intensities ranging from 0.5 to 10 mW, we either observed phototoxic induction of cell death at higher intensities (evident through ineffective erythromycin treatment) or ineffective induction of cell death at lower intensities. The used laser power, however, sufficed to bleach EGFP over the time course of 5 h, indicating that z-plane projection of light using a light sheet source is indeed possible, however, the compatibility likely requires additional efforts including homogenization of the light intensity throughout the tissue, and adjustments to better-suited wavelengths and tissue cultivation conditions.

## Optogenetic modulation of polarized WNT3A organizing centers

Developing tissues including embryos show a remarkable extent of self-organizing cell differentiation, resulting in morphological structures of high architectural order. The major underlying mechanisms during the formation and elaboration of structure-determining patterns in developing tissues have been identified as morphogen gradients, representing complex cell-cell communication networks involving the secretion of long-range-diffusible ligands and dose-dependent differentiated responses of receiving cells[60]. Organizer centers, the sources of local morphogen secretion, play a pivotal role in orchestrating gradients of cell-cell signaling molecules and give rise to various forms of pole formation. In vitro models, *e.g.*, reflecting embryonic development like stem cell-derived embryoid bodies (EBs), possess a much-attenuated propensity for self-organization compared with entire embryos, probably because of the artificial environment[61,62]. Therefore, tissue engineering seeks to develop synthetic organizing centers, assisting in establishing local and polarized signaling events. In order to best modulate how an engineered tissue differentiates and emerges, exogenously-controllable organizers are urgently needed to provide means to regulate the onset of polarized signals and thus guide developmental fates in vitro. An optimal interference strategy would allow modulating signaling events with a

patterned distribution, ultimately enabling precise adjustments of the strength, time, and location of developmental signals within tissues. Most currently used tissue engineering methods resort to bioprinting and microfluidics to control the spatiotemporal action of chemical signals. The combined use of optogenetic and genomic engineering approaches would constitute a general framework providing superior control capabilities. Therefore, we devised engineered optogenetic cells based on the BLUE_SINGLE vector system for modulating signaling events across 3D in vitro tissues.

Recently, we developed HEK^Cdh3-Wnt3a cells, which act as chemically-inducible organizers for mouse WNT3A secretion assisting in modulating gastrulation-stage early mouse embryonic development in EBs[63]. The delicate balance of *Wnt3*, *Bmp4*, and *Nodal* expression in the posterior, and long-range Wnt/β-catenin and Nodal/TGFβ inhibitors in the anterior visceral endoderm finely balance gastrulation-stage development in mouse embryos[64]. HEK^Cdh3-Wnt3a cells produce both mouse WNT3A and P-cadherin (CDH3) upon induction with tetracycline. We could show that induced HEK^Cdh3-Wnt3a cells self-segregate due to differential adhesion from cells expressing other cadherin types, such as E-cadherin (CDH1) present on mouse embryonic stem cells (ESCs), effectively forming self-organized three-dimensional centers which attach to EBs. Furthermore, the WNT3A secretion causes β-catenin signaling in proximal ESCs, resulting in the emergence of a T-brachyury-marked nascent mesoderm[63]. While the approach demonstrated a valuable method for maturing in vitro EB differentiation protocols, optogenetic methods would enable differential and sequential induction of individual organizers due to the spatial and temporal resolution and open avenues to precisely study processes, *e.g.*, underlying embryonal development aspects.

To achieve this goal, we engineered optogenetic gene expression switches for light-controlled *Wnt3a* expression in combination with chemically induced *Cdh3* expression into HEK-293 cells (Fig. 6A). Each of the two developed cell lines contains four independent genomically-integrated and differentially-selected vectors; two for tetracycline-regulated *Cdh3* expression utilizing the T-REx system[65], and two constituting either red or blue light-controlled expression of the *Wnt3a* gene utilizing RED_E or BLUE_SINGLE, respectively. The use of E as DNA-binding domain for promoter regulation served orthogonal control to T-REx which makes use of TetR to independently regulate the expression of *Cdh3* and *Wnt3a*.

We tested polyclonal HEK-293-derived cultures for red and blue light-induction of *Wnt3a* expression by quantitative RT-PCR (qPCR) specific to the mouse transgene (Fig. 6B). In cultures controlling *Wnt3a* expression by RED_E, we could not identify a clear induction of transcript levels under the tested conditions. This observation is in line with the data we obtained with HEK-293 cells for red light-induced SEAP expression, which showed only weak induction levels combined with a high basal expression (Fig. S2, Supplementary Information). However, for the polyclonal cell culture containing BLUE_SINGLE (HEK^Cdh3-OptoWnt), we observed an approximately 20-fold induction of transcript levels after 24 h of blue light illumination (Fig. 6B). In order

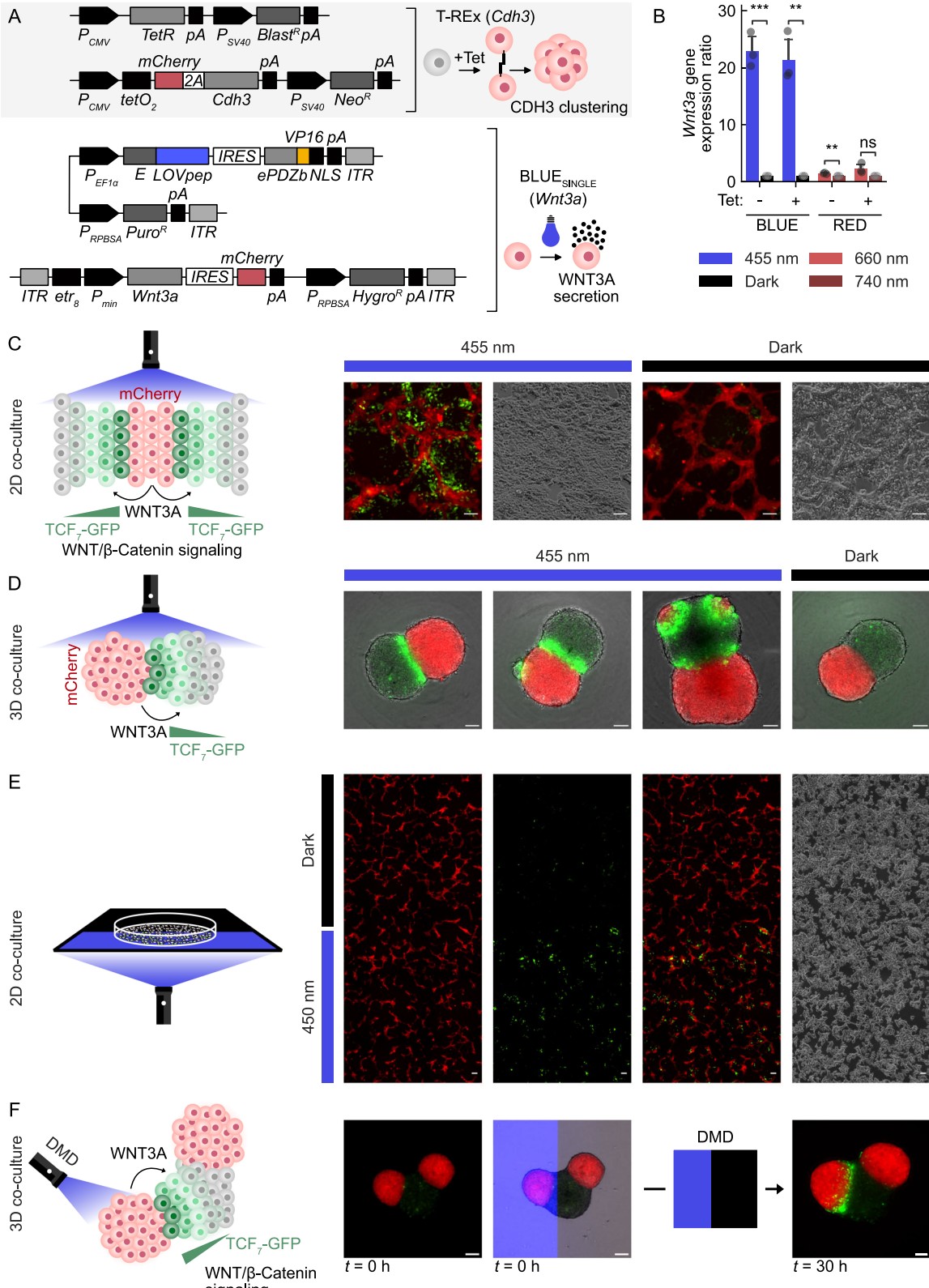

to test whether the cells would successfully secrete mature WNT3A into the extracellular space and activate Wnt/β-catenin in proximal tissue, we used a previously described HEK-293T-derived cell line (TOP-GFP) containing a synthetic β-catenin-specific *7xTCF/LEF* optimal promoter cassette (*7xTOP*) controlling *EGFP* expression in response to WNT3A signaling[66]. Mixing HEK^Cdh3-OptoWnt WNT3A sender with TOP-GFP receiver cells under tetracycline addition resulted in self-segregation

and the formation of discrete cell population patterns after cell adherence in 2D. HEK^Cdh3-OptoWnt formed dense web-like patches which could conveniently be identified due to the tet-induced *mCherry* expression, which were surrounded by islands of TOP-GFP cells (Fig. 6C). In blue light-illuminated cultures, WNT3A perception became evident in TOP-GFP cells proximal to HEK^Cdh3-OptoWnt patches, with EGFP fluorescence decreasing gradually with increasing distance. The local

**Fig. 6 | Optogenetic spatial modulation of polarized WNT3A organizing centers in 2D and 3D cultures. A** Engineering of HEK-293 cells for light-controlled expression of *Wnt3a* combined with chemically-inducible *Cdh3* expression (HEK[Cdh3-OptoWnt] cells). The engineered cell lines harbor four genomic constructs mediating the tetracycline-controlled expression of the mouse endothelial cadherin *Cdh3* for clustering and *mCherry* (top), and optogenetically-regulated expression and secretion of mouse *Wnt3a* using either a blue-light or red-light optogenetic gene-switch. **B** Quantitative RT-PCR analysis of optogenetic *Wnt3a* expression in HEK[Cdh3-OptoWnt] cells. HEK[Cdh3] cells with tetracycline-inducible *Cdh3* expression were further engineered for blue or red light-inducible expression of *Wnt3a* using the BLUE[SINGLE] or RED[E] system, respectively. Cells were illuminated with the corresponding wavelengths for 24 h prior to RNA extraction and qPCR analysis. Data represent mean values with one standard deviation of three technical replicates compared with two-sided independent Student's *t*-tests. ns, 5e-2 <*p* ≤ 1; **, 1e-3 <*p* ≤ 1e-2; ***, 1e-4 <*p* ≤ 1e-3. *p*-values: BLUE-Tet, 2.407e-4; BLUE+Tet, 1.287e-3; RED-Tet, 2.163e-3; RED+Tet, 6.885e-2. Source data are provided in this paper. **C** Optogenetic WNT3A-signaling across cell populations. HEK[Cdh3-OptoWnt] cells were induced with tetracycline for endothelial cadherin-mediated self-clustering and *mCherry* expression and mixed with TOP-GFP sensor cells for WNT3A perception and detection via *EGFP* expression. Cell cultures were illuminated for 24 h with 10 $\mu$mol m$^{-2}$ s$^{-1}$ blue light (overall illumination) using LEDs, or kept in darkness.

Representative images of four experiments are shown. Scale bar, 100 $\mu$m. **D** Three-dimensional optogenetic WNT3A organizing centers formed from HEK[Cdh3-OptoWnt] cells inducing WNT3A-signaling in proximal TOP-GFP spheroids. Assembloids were formed from 3000 or 6000 tetracycline-induced HEK[Cdh3-OptoWnt] cells and the same number of TOP-GFP cells were illuminated with 20 $\mu$mol m$^{-2}$ s$^{-1}$ blue light (overall illumination) using LEDs for 30 h, or kept in darkness. Representative images of one out of three similar experiments with 18 illuminated and four dark-incubated samples are shown. Scale bar, 100 $\mu$m. **E** Spatial induction of WNT3A signaling in a two-dimensional HEK[Cdh3-OptoWnt] and TOP-GFP co-culture using a photomask. Experiment as in (**C**) but the cells were illuminated from below through a 3D-printed photomask covering half of the sample area. A representative area of the acquired stitched wide-field region of a single experiment is shown. Scale bar, 100 $\mu$m. **F** Selective patterned activation of a distinct 3D optogenetic WNT3A organizing center and signaling perception by a TOP-GFP spheroid assembled with multiple organizers. Experiment as in (**D**) but using 2× 4000 HEK[Cdh3-OptoWnt] and 8000 TOP-GFP cells and digital mirror-controlled 450 nm blue light illumination at the microscope. A 4X objective was used for the initial image and pattern projection of a single selected sample which successfully formed a tripartite assembloid. The final image was acquired using a 10X objective. Scale bar, 100 $\mu$m. **A**, **B** Tet, tetracycline.

concentration of WNT3A-producing cells in this experiment combined with the TOP-GFP reporter cell line thus confirmed that bulk HEK[Cdh3-OptoWnt] are proficient in secreting functional WNT3A upon optical stimulation. We further tested if by selecting an individual high-expressing clone, overall WNT3A secretion could be enhanced (Fig. S9, Supplementary Information). Surprisingly, most of the clones were responsive and the polyclonal bulk culture performed comparably to the best ones, so we further used the bulk.

Next, we tested whether optogenetic induction could propagate WNT3A signals across merged three-dimensional tissues (Fig. 6D). We prepared spheroids of HEK[Cdh3-OptoWnt] and TOP-GFP cells followed by controlled fusion. We then illuminated the entire resulting assembloids with blue light and imaged EGFP fluorescence the next day. Bright EGFP signals emerging from the fusion sites indicated WNT3A secretion and reception by HEK[Cdh3-OptoWnt] and TOP-GFP 3D cultures, respectively. This suggests that three-dimensionally cultured HEK[Cdh3-OptoWnt] cells may indeed form functional synthetic and optogenetically controllable WNT3A organizers.

In contrast to existing *Wnt*-expression organizers, *e.g.*, those that act upon chemical stimulation, HEK[Cdh3-OptoWnt] cells provide means to induce Wnt secretion at desired spatial resolution in both two and three-dimensional tissue cultures. The several means of regulating light signals spatially can cover the spatial range from a single cell (laser) up to microscopic (DMD) and macroscopic (photomask, projection) scales. We first used a photo masking approach for specifying *Wnt3a* expressing and non-expressing zones in two-dimensional cultures (Fig. 6E). As expected, WNT3A-mediated activation of TOP-GFP cells occurred in blue light-illuminated regions rather than in those that were covered by the mask. We then resorted to the DMD device to explore the possibility of spatially activating one distinct WNT3A organizer within an assembloid tissue, where a TOP-GFP tissue possessed multiple organizers (Fig. 6F). We placed an illumination border within the microscopic view to explicitly expose just one organizer. This resulted in a sharp spatially-resolved differential activation of only this organizer, as indicated by EGFP-visualized WNT3A perception of the TOP-GFP tissue.

## Discussion

In most areas of technological development, enhanced accuracy and precision ultimately mark the progression from one generation of technology to the next. Manufacturing scales and tolerances of semiconductors determine the computational power of monolithic integrated circuits and finer scales are associated with higher quality,

performance, exchangeability, reliability, and added value for industrial products[67]. Similarly, finer-scale and higher precision technologies related to molecular biology, for example, DNA synthesis, and sequencing, have significantly contributed to the technological assets developed during the past decades. CRISPR/Cas-derived genome-engineering methods represent the highest form of accuracy that currently transitions into clinically relevant medical applications[68].

The concept of utilizing optogenetic technology for controlling cell function offers the potential to equip the biology of multidimensional in vitro tissues with greatly enhanced precision of regulation in space and time. The realization of this aim, however, requires an architectural implementation that fully supports in vitro tissues to gain precision control both in two and three dimensions. We approached the implementation by developing genomic engineering strategies paired with sophisticated illumination technologies and optogenetic gene switches that may be applied to a wide range of functional target genes, challenged by selecting molecular effectors requiring homogeneous light responsiveness among all tissue-resident cells. Optogenetically regulating the induction of various forms of programmed cell death, including apoptosis and necroptosis, requires all cells to be able to respond evenly to light stimuli. Such even response becomes possible through efficient switch designs as well as genome engineering and clonal selection protocols. The regulation of cell homeostasis and pattern elaboration represent major hallmarks of embryonic developing tissue. Gaining optogenetic control over such cross-tissue processes opens a new approach to studying these early processes in three-dimensional in vitro tissues. Our ability to control gene transcription in living organisms has for a long time been restricted to methods that entirely lack spatial precision and suffer from only limited temporal resolution, despite the high significance of inducible molecular switches for biomedical research and industry. While optogenetic solutions to enhance precision and accuracy for diverse biological studies are actively being developed and refined, including optogenetic gene switches[6], proof-of-concept experiments commonly rely on transiently introducing genetic components into organisms which might experience bottlenecks when applied in sophisticated experimental setups, *e.g.*, in vivo or in artificial multi-dimensional tissues which ask for a high and even spatial resolution.

In the present work, we systematically implemented variants of three optogenetic gene switches for their suitability in developing genomically-encoded artificial mammalian tissues that can elicit a strong optogenetic response. Utilizing the LOVpep/ePDZb (BLUE) and the PhyB[N]/PIF6[APB] (RED) gene switches combined with synthetic

Sleeping Beauty-based transposons enabled us to derive cell lines and in vitro tissues that we could precisely modulate using optical regulation. The approaches required highly tailored equipment to enable local sample illumination. We combined five major strategies to test the optogenetic gene switches. We applied (i) all-over illumination using LED light sources in early tests which we then (ii) combined with 3D-printed photomasks to enable spatial illumination regimes on the macroscopic scale. For high-resolution stimulation, we resorted either to (iii) pattern projection using a digital micromirror device connected to an epifluorescence microscope, or (iv) utilizing the excitation lasers of a confocal laser scanning unit. In addition, we (v) attempted to activate the optogenetic BLUE$_{SINGLE}$ switch in one plane by projecting a light sheet into 3D tissues. The latter three strategies enable highly precise, dynamic, and patterned activation simultaneously. We successfully utilized DMD and confocal laser activation for the optogenetic induction of cell death, however, our attempts in utilizing light sheets seem less compatible with the BLUE$_{SINGLE}$ photoswitch and likely require further optimization including a change of the light wavelength to reduce bleaching, and more sensitive optoswitches resulting in stronger optogenetic stimulation in 3D tissues. Red light-responsive photoswitches promise superior tissue penetration but often require exogenous cofactors, such as in the case of the PhyB/PIF-based RED systems. However, bacterial phytochromes relying on cell-available biliverdin might represent valid options that have recently been utilized in mammalian cell optogenetics[69,70]. Alternative variants of LOV2-domain-based optoswitches, including the improved light-induced dimers (iLIDs), exist in addition to the LOVpep/ePDZb (a.k.a. TULIP) system and represent a valid option for application optimization[71,72]. iLID Nano and Micro possess higher binding affinities and greater dynamic ranges between the dark and lit states as compared with the LOVpep system. However, the protein couples associate tighter or to a comparable extend in the dark state as the lid state of LOVpep, while iLID Milli proteins associate overall weaker, requiring appropriate evaluation and custom engineering strategies.

With vectors based on the LOVpep/ePDZb system, we successfully induced the expression of constitutive effectors of cell necroptosis both in two- and three-dimensional tissue cultures using blue and red light. Likewise, we succeeded in designing synthetic WNT3A-producing three-dimensional organizing centers capable of elaborating morphogen gradients with optogenetically-guided directionality across the boundaries of heterogeneous 2D and 3D in vitro tissues.

Conventional approaches like genetic modifications and chemical induction present significant experimental constraints for advanced 2D and 3D tissue applications. Optogenetics emerges as a viable solution to these obstacles. However, its application in 3D in vitro tissues has been limited, primarily due to the intricate requirements of accurately deploying optogenetic gene switches and the precise light stimulation needed, as well as the difficulties in maintaining long-term tissue cultures[32,73–75]. In this study, we integrated optogenetics with genome engineering, sophisticated biological circuits, and advanced illumination technology to develop robust, light-controllable systems for the next generation of tissue engineering. We anticipate that our findings on engineered optogenetic cells will catalyze the field, encouraging the creation of methodological frameworks to bring optogenetic tissues to the forefront of future research.

## Methods
### Molecular cloning
All plasmids constructed in this study were generated using standard restriction and PCR-based methods with Gibson[76], AQUA[77], and ligation cloning. Transposition-competent vectors encoding optogenetic switches were derived from plasmids pKM022[21] and pKM300[49] (RED, PhyB$_N$/PIF6$_{APB}$), pKM516[22] and pKM549 (BLUE, LOVpep/ePDZb), pIRES-Puro-VP-EL222[24] (EL222), and vector elements of pSBbi-GP (addgene #60511), pSBbi-RP (addgene #60513), and pSBtet-GP

(#60495). The effector and reporter genes $MLKL_N$, $Wnt3a$, and $SEAP$ were synthesized (IDT, Coralville, IA, US) or derived from plasmids pENTR-mCherry-2A-Wnt3a[63] or pKM006[21], respectively. Refer to Table S1, Supplementary Information, for a complete description, and to the Supplementary Data 1 file, Supplementary Information, for relevant vector maps.

### Cell culture and cell line engineering
Human embryonic kidney HEK-293 (DSMZ, Braunschweig, Germany, ACC 305), HEK-293T (DSMZ, ACC 635), HEK-293T-derived TOP-GFP[66], HEK$^{Cdh3}$ cells[63], and HeLa (DSMZ, ACC 57) cells and their derivatives were cultivated in a humidified atmosphere at 37 °C, 5% $CO_2$ in Dulbecco's Modified Eagle Medium (DMEM, PAN Biotech, Aidenbach, Germany, cat. no. P04-03550) supplemented with 125 U/mL penicillin, 125 U/mL streptomycin (PAN Biotech, cat. no. P06-07100), and 10% fetal bovine serum (PAN Biotech, cat. no. P30-3602). For CHO-K1 cells (DSMZ, ACC 110), Ham's F12 Medium (PAN Biotech, cat. no. P04-14500) with identical supplementation was used instead. For transient transfection, 50,000 cells (75,000 cells for HEK-293) were seeded per well into 24-well plates and transfected the next day using polyethyleneimine (PEI, linear, MW: 25 kDa, Polysciences Inc. Europe, Hirschberg, Germany, cat. no. 23966 (1)) as described elsewhere[33]. Briefly, PEI solution (1 mg/mL in $H_2O$) was adjusted to pH 7.0 with HCl, sterile filtered, and stored at −80 °C for further use. Per well, 0.75 µg of DNA were diluted in 50 µL of Opti-MEM (Thermo Fisher, Waltham, MA, US, cat. no. 22600134) and mixed with 2.5 µL of PEI solution diluted in 50 µL of Opti-MEM under mixing. After 15 min incubation at room temperature, the precipitate was added to the cells and the medium exchanged with fresh medium after 4 h. All plasmids were transfected in equal amounts (w:w), except for the experiment in Fig. 1G, where the plasmids encoding the photoswitches were used in 10-fold excess, the indicated conditions in Figs. 2 and S2, Supplementary Information, and when generating stable cell lines (see below).

For genomic integration of transposition-competent vectors, 250,000-350,000 cells of the cell line were seeded into 6-well plates and transfected with PEI as described above with the DNA amounts and volumes scaled up according to the growth area. For generating HEK$^{Cdh3-OptoWnt}$ cells, HEK$^{Cdh3}$ cells were used as the starting cell line[63]. Target vector DNA was mixed to contain 1/10 of plasmid pCMV(CAT)T7-SB100 (addgene #34879) encoding the SB100X transposase prior to transfection[46]. 24-48 h post-transfection, HEK-293, HEK-293T, HeLa, and CHO-K1 cells were put under antibiotic selection by the addition of 4, 10, 10, 10 µg/mL puromycin (Thermo Fisher, cat. no. A1113803), respectively, for the selection of photoswitch-encoding vectors. For reporter constructs, 400 µg/mL hygromycin, or 4, 10, 4, or 5 µg/mL blasticidin (InvivoGen, San Diego, CA, US cat. no. ant-hg-1, ant-bl-05), respectively, were used. The selection typically lasted for two to three weeks. Individual cell clones were randomly selected and cultivated without antibiotic selection. Polyclonal cultures and individual clones were cryopreserved in liquid nitrogen.

For microscopic analysis of necroptosis as in Figs. 4B and S3–S4, Supplementary Information, 200,000 cells were seeded in 6-well vessels one day prior to light induction either in light boxes or via a microscopic setup. To protect cells from undergoing light-induced necroptosis, the cultures were supplemented with 2 µg/mL erythromycin (Sigma Aldrich, St. Louis, MO, US, cat no. E5389-1G). HEK$^{Cdh3-OptoWnt}$ and TOP-GFP cells were mixed in 6-wells at a density of 500,000 cells, each, and supplemented with 1 µg/mL tetracycline (Sigma Aldrich, cat. no. T7660-5G). In experiments involving the red light photoreceptor PhyB, cultures were supplemented with 15 µM phycocyanobilin 1 h before illumination (PCB, Frontier Scientific, Logan, UT, US, cat. no. P14137-50mg). Cell death was visualized with 1 µM SYTOX Blue (Thermo Fisher, cat. no. S11348).

Three-dimensional spheroid cultures were generated using round bottom BIOFLOAT-coated 96-well plates (faCellitate, Mannheim,

Germany, cat. no. F202003) with cell suspensions of 100 μL containing the indicated cell numbers and the above-described supplements. HEK$^{Cdh3-OptoWnt}$ cells were additionally supplemented with 2 μg/mL tetracycline. After overnight incubation, spheroids were mounted in a medium containing 1% low-gelling agarose (Sigma Aldrich, cat. no. A4018-5G) for live imaging, or first merged into assembloids by pipetting using cut tips and additional incubation for 12–24 h. For light sheet microscopy, spheroid cultures were mounted in 1% low-gelling agarose (Sigma Aldrich, cat. no. A9414-5G) dissolved in PBS, and pulled up into glass capillaries (Sigma Aldrich, cat. no. Z328480).

### Sample illumination

Experiments were performed with the indicated light intensities. LED light sources were calibrated using an AvaSpec-ULS2048-USB2 fiber optic spectrometer (Avantes, Apeldoorn, Netherlands) connected to an FC-UVIR200-2-ME-FC/SMA sensor head. All-over illumination of cell cultures was performed in tailor-made light boxes housing blue (455 nm), red (660 nm), or far-red (740 nm) LEDs. For spatial illumination, photomasks were 3D-printed and samples were illuminated from below either using the light boxes (HEK$^{Cdh3-OptoWnt}$, 10 μmol m$^{-2}$ s$^{-1}$ 455 nm light), or the Light Plate Apparatus (LPA)[58] equipped with 460 nm LEDs (Mouser, Mansfield, TA, US, cat. no. VAOL-5GSBY4) using apparent light intensities ranging between 20 and 70 μmol m$^{-2}$ s$^{-1}$. Dose-response curves were obtained by illumination of cells with the indicated light intensities using LPA[58] units that were customized with a 3D-printed chassis and two layers of Makrofol LM 309-2-4 diffusor foils (Covestro AG, Leverkusen, Germany) for homogeneous illumination. The LPAs were equipped with blue 445 – 465 nm blue light LEDs (Mouser, Mansfield, TA, US, cat. no. 1124_C503B_BCS_BCN_GCS_GCN_030-2326560, or Roithner Lasertechnik GmbH, Vienna, Austria, cat. no. led450-03).

For spatial illumination during live cell imaging, a custom-designed digital mirror device was used connected to a 450 nm mounted LED (Thorlabs, Newton, NJ, U, cat. no. M450LP1) projected through a CFI Plan Apochromat λ 4x/NA = 0.20 objective (Nikon, Minato City, Tokyo, Japan) attached to a Ti2 microscope (Nikon). The light intensity at the sample plane was set to 30 μmol m$^{-2}$ s$^{-1}$. However, this value serves data reproducibility and does not reflect the true intensity, as the projection field does not cover the entire sensor surface. Details on the DMD device are described elsewhere[59]. Laser stimulation using the confocal microscope was done through a CFI Plan Fluor 10X/NA = 0.30 objective (Nikon) using the 488 nm laser of a LU-N4 laser unit (Nikon) in photo-stimulation mode.

In light sheet microscopy (see description of the microscope in *Imaging*), we used the light sheet itself to illuminate one plane in spheroids (488 nm laser wavelength, illumination only from one side). The light sheet thickness was ~5 μm crossing the entire spheroid along the lateral axis. The laser output power varied between 0.5 and 10 mW. To visualize necroptotic cells, cultures were supplemented with 1 μM SYTOX Blue or 5 nM SYTOX Red Nucleic Acid Stain (Thermo Fisher, cat no. S11348, S34859).

### Analysis of gene expression

SEAP reporter levels were determined from cell culture supernatants as described elsewhere[23]. In brief, cells were seeded and transfected the next day. Illumination was started 24 h post-transfection. Stable cells rested for one day prior to illumination and the medium was exchanged. At the end of the experiment, 200 μL supernatant was harvested from each individual cell culture and incubated for 45 min in a heat oven at 65 °C. Samples were diluted in 80 μL PBS to obtain appropriate absorbance readings and added to 100 μL 2x concentrated assay buffer (20 mM homoarginine (Thermo Fisher, cat. no. H27387.14), 1 mM MgCl$_2$, 21% (w/v) diethanolamine/HCl (pH 9.8)). The reaction was started by the addition of 20 μL of 120 mM paranitrophenyl phosphate (Sigma Aldrich cat. no. 71768-25 G) solution dissolved in H$_2$O. Absorbance readings at 405 nm were followed for 1 h using a CLARIOstar (BMG LABTECH, Ortenberg, Germany) or TriStar2S (Berthold Technologies, Bad Wildbad, Germany) plate reader.

Quantitative RT-PCR was performed using the GoTaq qPCR master mix (Promega, Madison, WI, US, cat. no. A6001) in a StepOnePlus real-time PCR system (Applied Biosystems, Foster City, CA, US) with 10 ng of cDNA per sample in triplicate reactions according to the instructions of the manufacturers. RNA was isolated using the NucleoSpin RNA Plus Kit (Macherey Nagel, Düren, Germany, cat. no. 740984.50) and converted into cDNA using the LunaScript RT SuperMix Kit (NEB, Ipswich, MA, US, cat. no. M3010L). Mouse *Wnt3a* was probed using the oligonucleotides oDD337 (5′-CATGCACCTCAAGTGCAAATGC) and oDD338 (5′-TGAGGAAATCCCCGATGGTG) derived from a previously published sequence[78]. *GAPDH* was detected with oDD145 (5′-GTCTCCTCTGACTTCAACAGCG) and oDD146 (5′-ACCACCCTGTTGCTGTAGCCAA) with the sequence derived from GAPDH Human qPCR Primer Pair (Origene Technologies, Rockville, MD, US, cat. no. HP205798). qPCR data were analyzed as relative gene expression data according to the Pfaffl method[79].

### Imaging

Microscopic imaging was performed using a Ti2 inverted microscope (Nikon) equipped with a C2 confocal and LU-N4 laser unit (Nikon) equipped with 405 and 488 nm lasers for SYTOX Blue and EGFP excitation, respectively. 475/50 and 535/30 nm emission filters and a CFI Plan Fluor 10X/NA = 0.30 objective (Nikon) were used. Epifluorescence images were acquired using an ORCA-Flash4.0 V2 Digital CMOS camera (Hamamatsu, Hamamatsu City, Shizuoka, Japan) with a CFI Plan Apochromat λ 4x/NA = 0.20 or a CFI Plan Fluor 10X/NA = 0.30 objective (Nikon). SYTOX Blue, EGFP, and mCherry were imaged using 434/17, 470/40, 578/21 nm excitation, and 479/40, 525/50, 641/75 nm emission filters, respectively. Brightness and contrast were adjusted using ImageJ 2.3.0/1.5.3q (NIH, Bethesda, MD, US). Light sheet imaging was performed using a custom-built microscope featuring 488 nm and 647 nm LuxX diode lasers (Omicron-Laserage Laserprodukte GmbH, Rodgau-Dudenhofen, Germany), cylindrical lenses to create the light sheet, dual-sided illumination via two 10x/0.2 objectives (Carl Zeiss AG, Feldbach, Germany), single-sided detection through a 16x/0.8 water dipping objective (CFI75 LWD 16X W), and an sCMOS camera (Iris 15, Teledyne Scientific Imaging GMBH, Surrey, British Columbia, Canada). The sample was moved in xyz and rotated using motorized stages (Physik Instrumente (PI) GmbH & Co. KG, Karlsruhe, Germany). EGFP and SYTOX Red were imaged using 525/50 (Chroma Technology Corporation, Bellow Falls, USA) and 731/137 (Semrock, IDEX Health & Science LLC, Rochester, USA) emission filters, respectively. After one-sided optogenetic illumination of one plane for 5 h, every spheroid was imaged from two angles (0 and 180°) and sequentially illuminated from both sides. The four resulting 3D stacks were fused and deconvolved using Huygens (Scientific Volume Imaging B.V., Hilversum, Netherlands).

### Dose-response curves

Dose-response experiments in Fig. 3E (quantification of SEAP production) and Fig. 4C (quantification of cell survival) were performed by seeding 100,000 CHO-K1 BLUE$_{SINGLE}$ #14, or 25,000 CHO-K1$^{NecrOpto}$ cells into 24-wells. The next day, the cells were illuminated as described above. SEAP production was subsequently analyzed as described above. For determining cell survival, the adherent cultures were washed three times with PBS to remove necroptotic cells. For each sample, a 4×2 stitched image area of 11.75 mm$^2$ was acquired using a 10X objective, and the number of EGFP positive cells was counted using cell segmentation with CellProfiler 4.2.6 (Broad Institute, Massachusetts Institute of Technology, MA)[80]. Data were fitted into the Hill isotherm models:

$S(i) = \text{p0} + \dfrac{\text{p1}}{1 + \left(\frac{I^{50}}{i}\right)^n}$ for the SEAP quantification, and

$C(i) = \text{p0} + \text{p1}\left(1 - \dfrac{1}{1 + \left(\frac{I^{50}}{i}\right)^n}\right)$ for the quantification of cell survival using

Python 3.11 and the scipy 1.13.1 package, where $I^{50}$ is the light intensity at half-maximal response and $n$ is the Hill coefficient.

## Statistical analysis

Optogenetic gene expression experiments using the SEAP gene were performed in four biological replicates either representing cultures of a cell line or originating from individual plasmid transfections. For the 455 nm light sample in Fig. 2C with 1/10 of the SEAP reporter constructs, and the 660 nm light sample in HEK-293T cells with C,R$_t$ in Fig. S2B, Supplementary Information, three values were used due to a pipetting error. Plotted data represent mean values with one standard deviation calculated with Python 3.11 and the seaborn 0.11.2 package, overlaid with single data points. The qPCR analysis was performed in triplicates. Two-sided independent Student's $t$-tests were performed using Python 3.11 and the statannotations 0.6.0 and scipy 1.13.1 packages with ns, $5\text{e-}2 < p \le 1$; *, $1\text{e-}2 < p \le 5\text{e-}2$; **, $1\text{e-}3 < p \le 1\text{e-}2$; ***, $1\text{e-}4 < p \le 1\text{e-}3$; ****, $p \le 1\text{e-}4$.

## Reporting summary

Further information on research design is available in the Nature Portfolio Reporting Summary linked to this article.

## Data availability

All data generated in this study are provided in the Supplementary Information, the Source Data file, and the Supplementary files contained in the archive Supplementary Data 1.zip. Source data are provided with this paper. Materials are available on request. Source data are provided with this paper.

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

## Acknowledgements

We thank F. Decker (ETH, Zürich), S. Kuschel, and R. Schönle (Düsseldorf, Germany) for valuable experimental assistance. We are grateful to A. Geidies, C. Fischer, and A. Teixeira (Düsseldorf, Germany) for advice on data analysis, and to K. Müller and W. Weber (Freiburg, Germany) for useful DNA constructs listed in Table S1, Supplementary Information. We are grateful to G. Davidson (Karlsruhe, Germany) for providing the TOP-GFP cell line. The project was supported by the *Freigeist* Fellowship of the Volkswagen Foundation to H.M.B., and the Deutsche Foschungs-gemeinschaft (DFG, German Research Foundation) under the Germany´s Excellence Strategy (EXC-2048 project no. 390686111), NEXTPlant (Project ID 391465903/GRK 2466), the Collaborative Research Centers SFB1208 (project no. 267205415) and SFB1535 (project no. 458090666), the Human Frontiers Scientific Program HFSP (HFSP RGY0063/2017 and RGP 0067/2021 Grants) to M.D.Z., and the European Commission – Research Executive Agency (H2020 Future and Emerging Technologies (FET-Open) Project ID 801041 CyGenTiG to H.M.B., S.K., K.T., C.F., J.A.D., M.K., and M.D.Z.

## Author contributions

M.K. and M.D.Z. conceived the project, secured funding, and supervised the project. H.M.B. secured funding, designed, supervised, and performed experiments, and analyzed data. S.K. developed the DMD platform and performed spatial necroptosis experiments. M.N. established and performed the confocal microscopy and photo masking experiments, and generated necroptosis cells. C.M.C.D., K.T., S.S., C.S.K. designed and tested optogenetic DNA constructs. H.M.B., M.K., and M.D.Z. coordinated the study. C.F., J.A.D., M.K., and M.D.Z. provided infrastructure. H.M.B. wrote the manuscript. The final version contained input from all authors.

## Funding

## Competing interests

The authors declare no competing interests.
