## [Transparent Peer Review file · Nature Communications]

Genetically-stable engineered optogenetic gene switches modulate spatial cell morphogenesis in two- and three-dimensional tissue cultures

Corresponding Author: Professor Matias Zurbriggen

Version 0:

Reviewer comments:

Reviewer #1

(Remarks to the Author)

The manuscript by Beyer et al. presents a detailed study of the optogenetic control of gene transcription. First, this methodological work compares the efficiency of transcription induction for different optogenetic systems, all based on the dimerization of a split transcription factor (heterodimerization for the red system and the single/dual blue systems, and homodimerization for EL222 blue system). The author compares different promoter architectures, transient transfection versus partial genomic integration and complete genomic integration (bulk and clones), and cell lines. This is an impressive amount of work, and the conclusion is that the blue single optogenetic switch works the best and that clonal selection is required. Second, the authors apply their optogenetic switches to two biological processes, namely necroptosis and organizing centers in embryoids. They demonstrate that it is possible to control those processes in 2D and 3D (albeit not with light sheet microscopy) with an excellent spatial control. The induction of organizing centers in 3D (Fig6D and F) is particularly impressive. Altogether, this work presents a relevant benchmarking study of optogenetic tools for gene transcription that will be useful for the community, with promising results on the control of biological processes, even if there is no new biological insight being presented.

The manuscript is very dense, and sometime hard to read, especially in the first part where multiple constructs are compared. It requires a quite important effort for the reader to follow the meaning of these constructs (P, R, Rt, Ra, etc.). Maybe the fig1 could be slightly improved so that the desired effect of each construct is explicitly stated in the figure (eg in 1E)? Or the labeling made more explicit (it was not clear what R, Rt and Ra meant before reading the caption)? To lighten the manuscript in the second part, the section describing the DMD systems could be removed since it is extensively presented in a second manuscript.

Regarding the choice of the optogenetic switches, it would be good to discuss a little bit more why the other tools such as or clustering or protein localization were not considered. There is just the mention to the ref 21 and 22 but there is no comparative analysis in these references.

Still with regards to the choice made in this work, I was wondering why the TULIP system (LOV2/ ePDZb) has been chosen, given that the system has been improved with the iLID/SSPB system. For example, in Fig1 of 10.1021/acssynbio.5b00119 it is shown that the dynamic range of iLID/SSPB is much higher than TULIP, thus I would expect it to be a much better system for induction of gene transcription (with larger fold increase and tighter basal expression level).

On the second part, given that the present work is mostly methodological, it would have been great to have dose-response curves of the biological outcome -at least in 2D. Indeed, for users it is always good to know what is the minimal/saturating amount of light and how it depends on the total exposure time. A quantification of the induction of necroptosis versus integral illumination would be useful.

Minor remark: there is a big black box on figure S5, should be removed.

Reviewer #2

(Remarks to the Author)

In this manuscript, the authors engineer light-sensitive mammalian cell lines by using light-regulated promoters, sensitive to red/far red and blue. The promoters were integrated using the sleeping beauty 100x transposase, selecting for the most efficient clone after confirming the integration.

The authors then proceed to demonstrate how the system they developed allows for inducing necroptosis and, in another experiment, for modulating WNT3A organizing centers. Several illumination techniques were used, ranging from widefield illumination to patterned illumination using both a mask and a DMD projector, a laser scanning confocal microscope and a lightsheet microscope.

The strategy for integrating the light-sensitive promoters in the genome of the mammalian cells is relatively interesting, albeit not novel, and allows for inserting a longer payload when compared to e.g. lentivirus infection based methods. The proof of concepts illustrated in the manuscript are generally adequate for the 2D case, but the manuscript fails to convincingly demonstrate a real three-dimensional induction of e.g. cell death (the images shown in Fig. 5 panel D is supposedly a max intensity projection and the sample was only illuminated at the periphery, while panel F shows a spheroid which seems to be illuminated only at its outer surface by the lightsheet). Same more interesting results are achieved for the modulation of WNT3A organising centres, unfortunately only using 2D illumination.

In summary, the manuscript describes a method for creating light-responsive cell lines and a validation strategy. While the work demonstrates manipulation of existing systems to suit the authors' requirements, in my opinion lacks novelty and I would recommend publication in a more specific journal.

Minor comments

The abstract describe two- and three-dimensional optogenetic patterning of tissue, but the manuscript focuses on the genetic methods used to integrate the light-sensitive promoter in the genome.

Figure 1 panel F: are the +Act and -Act inverted?

Line 801: 0.2 a.u. of laser intensity is meaningless, either remove the value or give an estimate

Lines 804-805: the reported lightsheet thickness is 5 μm is said to "cover the entire spheroid", which I believe refers to the lateral extent of the lightsheet. The authors should better clarify the geometry of the lightsheet illumination (the spheroid looks much larger than 100 μm). Also, the arrow in Figure 5 panel F and the scheme in panel E seem not to match with the actual propagation of the illumination, which I believe is from top to bottom in panel F.

Reviewer #3

(Remarks to the Author)

This is a very well planned and executed study. Even so it is a very big study it is questionable if the study should be published in two papers as it makes it much more difficult for the reader to follow the main points of the study as they get diluted. The reviewer suggests to combine two manuscripts into one to make the story line and main points more prominent so reader can easily follow and then extend the supplementary section extensively with the details not published in the main manuscript. This study is very well planned and executed. However, despite its size, it is questionable whether it should be published in two papers, as it could make it difficult for readers to follow the main points of the study. The reviewer suggests combining the two manuscripts into one to make the storyline and main points more prominent, enabling readers to easily follow them. The details not published in the main manuscript can then be extended in the supplementary section.

Version 1:

Reviewer comments:

Reviewer #1

(Remarks to the Author)

I thank the authors for their revised manuscript which answer all my requests.

Reviewer #2

(Remarks to the Author)

In the rebuttal letter, the authors address the points I have raised in my review and add further information which help clarifying the experimental arrangement used in the manuscript. Although the experimental part fails to achieve a convincing 3D induction of necroptosis, the biological system the authors have developed is a valid resource to test optogenetics.

REVIEWER COMMENTS

We appreciate the editor's evaluation of our work for further consideration, and we are thankful to the three reviewers for their assessments and valuable feedback. Their suggestions have strengthened our manuscript and enhanced its clarity. Based on the reviewers' comments, relevant modifications have been incorporated in the revised manuscript. These changes are highlighted in the manuscript text.

Below are the reviewers' comments and our point-by-point response (highlighted in blue).

Reviewer #1 (Remarks to the Author):

The manuscript by Beyer et al. presents a detailed study of the optogenetic control of gene transcription. First, this methodological work compares the efficiency of transcription induction for different optogenetic systems, all based on the dimerization of a split transcription factor (heterodimerization for the red system and the single/dual blue systems, and homodimerization for EL222 blue system). The author compares different promoter architectures, transient transfection versus partial genomic integration and complete genomic integration (bulk and clones), and cell lines. This is an impressive amount of work, and the conclusion is that the blue single optogenetic switch works the best and that clonal selection is required. Second, the authors apply their optogenetic switches to two biological processes, namely necroptosis and organizing centers in embryoids. They demonstrate that it is possible to control those processes in 2D and 3D (albeit not with light sheet microscopy) with an excellent spatial control. The induction of organizing centers in 3D (Fig6D and F) is particularly impressive. Altogether, this work presents a relevant benchmarking study of optogenetic tools for gene transcription that will be useful for the community, with promising results on the control of biological processes, even if there is no new biological insight being presented.

Response: We would like to thank Reviewer #1 for critically reviewing our work and the useful suggestions for improvement. We are delighted about the positive feedback and have carefully addressed all issues below.

Regarding our unsuccessful attempts in regulating optogenetic tissues with light sheet microscopy, we have now modified the abstract and do not mention light sheet microscopy any longer, so as not to evoke the wrong impression that we were successful with our approach. Nevertheless, we would like to keep the data in the manuscript to share our attempts to evaluate several of the illumination approaches and guide with our experience future tests by colleagues.

The manuscript is very dense, and sometime hard to read, especially in the first part where multiple constructs are compared. It requires a quite important effort for the reader to follow the meaning of these constructs (P, R, Rt, Ra, etc.). Maybe the fig1 could be slightly improved so that the desired effect of each construct is explicitly stated in the figure (eg in 1E?)? Or the labeling made more explicit (it was not clear what R, Rt and Ra meant before reading the caption)? To lighten the manuscript in the second part, the section describing the DMD systems could be removed since it is extensively presented in a second manuscript.

Response: We thank the reviewer for pointing out the issue. We agree that Figure 1 might be difficult to understand initially, requiring the figure legend for explanation. We now explicitly write out the meaning of the short names in Figure 1E as suggested. We also added additional labels to Figure 1D. We believe that the new version is now easier to understand.

In addition, we have removed the technical description of the DMD and only mention the working concept required for a general understanding of the technology with reference to the second manuscript (line 449):

“To select distinct culture areas for illumination and to gain insights into the kinetics of spatial necroptosis induction through modulation of light intensities combined with live cell imaging, we utilized our custom-designed DMD (Digital Micromirror Device)-based platform – μ PatternScope⁵⁹. This platform, when optically coupled with a microscope and a light source, defines high-resolution LED blue light patterns (1920x1080 micropixels) reflected from a DMD chip and focuses them onto the sample plane of the microscope.”

Regarding the choice of the optogenetic switches, it would be good to discuss a little bit more why the other tools such as or clustering or protein localization were not considered. There is just the mention to the ref 21 and 22 but there is no comparative analysis in these references.

Response: We thank the reviewer for pointing out the need for discussion regarding the choices of optogenetic switches tested in this work. We chose the included systems primarily based on our own experience on which would be adequate in terms of quantitative outputs for evaluating the spatiotemporal control capabilities. We have tested TULIP-, PhyB- and EL222-based systems lab internally using transient transfection approaches and also in other organisms over several years for various aims and projects, leaving us with some experience regarding their performance. We modified the respective section in the text to reflect this (line 134):

“We focused on variants of red/far-red and blue light-inducible gene expression systems that rely on the induction of protein heterodimerization. We concentrated here on systems we have developed or tested over the past years using transient transfection approaches and provide us with an easy to estimate quantitative output. In particular we selected those for which we would expect either good dynamic induction ranges²², reversibility²¹, or low basal expression levels based on our experience (Figure 1A-B). Additionally, we included a single-component blue light system relying on light-regulated modulation of DNA binding, because it has a simpler architecture and fewer required components (Figure 1C).”

Still with regards to the choice made in this work, I was wondering why the TULIP system (LOV2/ ePDZb) has been chosen, given that the system has been improved with the iLID/SSPB system. For example, in Fig1 of 10.1021/acssynbio.5b00119 it is shown that the dynamic range of ilid/SSPB is much higher than TULIP, thus I would expect it to be a much better system for induction of gene transcription (with larger fold increase and tighter basal expression level).

Response: The choice of the TULIP system primarily goes back to the fact that we build upon our previous work engaging a similar variant of the TULIP switch in transient cell transfection experiments (see 10.1021/sb500305v). As mentioned above, we have used this system ever since in various experiments with mainly satisfying results, achieving strong induction with light and low activation in the dark (suitable when controlling cell death factors), which made us eager to test TULIPs also in stably-engineered cells and with additional chemical regulation. As the TULIP and the iLID/SspB system both are based on the same photoreceptor, they fell into the same category in our tests and we did not consider switching to iLID.

However, we agree with the reviewer's points that the iLID/SspB systems with different binding affinities and dynamic ranges could serve as a strategy to further modulate the properties of the used gene expression architecture. This could particularly be useful for target genes requiring modifications in the expression strength or light response profile compared to the TULIP system, because according to 10.1021/acssynbio.5b00119, the dark states of iLIDs show stronger/comparable binding affinities to SspBnano/micro as ePDZ to LOVpep in the lit state. To benefit from a potentially larger fold increase without risking elevated basal

expression levels, a thorough refinement of the target promoter architecture likely would have to be accompanied by the redesign of the system. We have therefore expanded the discussion to point the readers to the improved iLID systems as a valid option for further optimization and target adjustment together with a discussion of red light switches. We might also consider the use of iLIDs in future projects e.g. where stronger transgene expression as compared to what is needed for cell death factors would be required, as we have yet to gain experience with them (line 718):

“Red light-responsive photoswitches promise superior tissue penetration but often require exogenous cofactors, such as in the case of the PhyB/PIF-based RED systems. However, bacterial phytochromes relying on cell available biliverdin might represent valid options that have recently been utilized in mammalian cell optogenetics^{69,70}. Alternative variants of LOV2-domain-based optoswitches including the improved light-induced dimers (iLIDs), exist in addition to the LOVpep/ePDZb (a.k.a. TULIP) system and represent a valid option for application optimization^{71,72}. iLID Nano and Micro possess higher binding affinities and greater dynamic ranges between the dark and lit states as compared with the LOVpep system. However, the protein couples associate tighter or to a comparable extend in the dark state as the lid state of LOVpep, while iLID Milli proteins associate overall weaker, requiring appropriate evaluation and custom engineering strategies.”

On the second part, given that the present work is mostly methodological, it would have great to have dose-response curves of the biological outcome -at least in 2D. Indeed, for users it always good to know what is the minimal/saturating amount of light and how it depends on the total exposure time. A quantification of the induction of necroptosis versus integral illumination would be useful.

Response: We agree with this point, the light intensity is indeed an important parameter. We have now added dose-response curves both for the BLUE_{SINGLE} cell clone #14 with the SEAP reporter and the CHO-K1^{NecrOpto} cell line where we quantified the extent of necroptosis by determining the number of living cells.

Due to the relatively small projection area, the sensor area of our photospectrometer is incompatible with DMD projection and it was impossible to accurately measure blue light intensities in absolute units at the focus plane at the microscope. Therefore, we used Light Plate Apparatus (LPA) units instead. We modified the devices by extending the light path length with a customized 3D printed chassis that has the well-separating walls removed and allowed us to install two diffusor films at different path lengths to obtain a very homogeneous illumination area and with that accurate dose-response curves. The blue light intensities we used ranged from 0 – 14 $\mu\text{mol s}^{-1} \text{m}^{-2}$ and included eight different intensity values. We determined I^{50} intensity values with half-maximal SEAP secretion or cell-survival of 6.45 and 3.03 $\mu\text{mol s}^{-1} \text{m}^{-2}$, respectively. The results are included as Figures 3E and 4C along with a discussion in the text and a description of the methods.

Minor remark: there is a big black box on figure S5, should be removed.

Response: Thanks for the observation. We found the box only after the submission as a conversion artifact. We have now changed the upload file format from *.DOCX to *.PDF to solve the issue.

Reviewer #2 (Remarks to the Author):

In this manuscript, the authors engineer light-sensitive mammalian cell lines by using light-regulated promoters, sensitive to red/far red and blue. The promoters were integrated using the sleeping beauty 100x transposase, selecting for the most efficient clone after confirming the integration.

The authors then proceed to demonstrate how the system they developed allows for inducing necroptosis and, in another experiment, for modulating WNT3A organizing centers. Several illumination techniques were used, ranging from widefield illumination to patterned illumination using both a mask and a DMD projector, a laser scanning confocal microscope and a lightsheet microscope.

The strategy for integrating the light-sensitive promoters in the genome of the mammalian cells is relatively interesting, albeit not novel, and allows for inserting a longer payload when compared to e.g. lentivirus infection based methods. The proof of concepts illustrated in the manuscript are generally adequate for the 2D case, but the manuscript fails to convincingly demonstrate a real three-dimensional induction of e.g. cell death (the images shown in Fig. 5 panel D is supposedly a max intensity projection and the sample was only illuminated at the periphery, while panel F shows a spheroid which seems to be illuminated only at its outer surface by the lightsheet). Same more interesting results are achieved for the modulation of WNT3A organising centres, unfortunately only using 2D illumination.

In summary, the manuscript describes a method for creating light-responsive cell lines and a validation strategy. While the work demonstrates manipulation of existing systems to suit the authors' requirements, in my opinion lacks novelty and I would recommend publication in a more specific journal.

Response: We would like to thank the reviewer for thoroughly reviewing our manuscript and her/his useful feedback.

In the revised manuscript, we added an additional supporting figure (Figure S8) which shows the individual layers of the intensity projection mentioned above in order to provide more insights into the depth of necroptosis induction in the experiment in Fig. 5D.

Regarding the light-sheet experiment in Fig. 5F, as stated in the manuscript, we were indeed unable to induce necroptosis in 3D with the desired precision. However, we still wanted to include the data and share our experience with interested readers who might aim for a similar optogenetic induction strategy.

In Fig. 6F we used the DMD device to induce WNT3A secretion upon optogenetic induction in a 3D assembloid tissue. While the illumination strategy can be considered both 2D or 3D depending on the interpretation, the biological effect engages morphogen secretion and diffusion. As we induced WNT3A secretion from and to three-dimensional tissues, the observed biological effect is in fact 3D.

Minor comments

The abstract describe two- and three-dimensional optogenetic patterning of tissue, but the manuscript focuses on the genetic methods used to integrate the light-sensitive promoter in the genome.

Response: We thank the reviewer for highlighting this point. The article deals with both the engineering and the implementation of advanced optogenetic methods for the spatiotemporal control in 2D and 3D of cellular processes. We have slightly changed the the title of the manuscript in order to better highlight the genetic engineering methods. (The 'engineering' componend is further highlighted by the short tile): "*Genetically-stable engineered optogenetic*

gene switches modulate spatial cell morphogenesis in two- and three-dimensional tissue cultures”

Figure 1 panel F: are the +Act and -Act inverted?

Response: Thank you very much, we have corrected the label.

Line 801: 0.2 a.u. of laser intensity is meaningless, either remove the value or give an estimate

Response: We have now removed the intensity statement.

Lines 804-805: the reported lightsheet thickness is 5 μm is said to "cover the entire spheroid", which I believe refers to the lateral extent of the lightsheet. The authors should better clarify the geometry of the lightsheet illumination (the spheroid looks much larger than 100 μm). Also, the arrow in Figure 5 panel F and the scheme in panel E seem not to match with the actual propagation of the illumination, which I believe is from top to bottom in panel F.

Response: We changed the description of the light sheet geometry relative to the spheroid as follows (lines 831):

“In light sheet microscopy (see description of the microscope in Imaging), we used the light sheet itself to illuminate one plane in spheroids (488 nm laser wavelength, illumination only from one side). The light sheet thickness was $\sim 5 \mu\text{m}$ crossing the entire spheroid along the lateral axis.”

The arrow in Figure 5, however, is correct as shown. The light sheet crosses the spheroids in the lateral plane with the position and direction indicated by the arrow.

Reviewer #3 (Remarks to the Author):

This is a very well planned and executed study. Even so it is a very big study it is questionable if the study should be published in two papers as it makes it much more difficult for the reader to follow the main points of the study as they get diluted.. The reviewer suggests to combine two manuscripts into one to make the story line and main [points more prominent so reader can easily follow and then extend the supplementary section extensively with the details not published in the main manuscript. This study is very well planned and executed. However, despite its size, it is questionable whether it should be published in two papers, as it could make it difficult for readers to follow the main points of the study. The reviewer suggests combining the two manuscripts into one to make the storyline and main points more prominent, enabling readers to easily follow them. The details not published in the main manuscript can then be extended in the supplementary section.

Response: We thank the reviewer for examining both manuscripts and for providing a favorable assessment. We decided to draft two separate manuscripts, as we see two distinct areas of interest for the community in our work. While one manuscript deals with the genomic engineering of cells to equip them with optogenetic gene switches and explores their potential to control 2D and 3D cellular processes, the other manuscript centers on the development and integration of the $\mu\text{PatternScope}$ hardware and software into optogenetic biological experiments.

The valuable feedback received from the reviewers during the revision process strengthens our strategy. Reviewer #1 of the $\mu\text{PatternScope}$ manuscript identified her/himself as a 'targeted reader' likely to implement a similar system, and it could be well possible that she/he aims to extend the utility of the device beyond mammalian cell systems. This feedback – as

we think – shows that there is a striking relevance of our developed illumination technology and that it generates interest within the target audience. Conversely, the primary readership of this manuscript may not have a particular interest in the technical details of μ PatternScope (in fact, Reviewer #1 suggested removing technical DMD description unrelated to the biological implementation, which we did). Our focus remains here on developing gene switches compatible with diverse illumination strategies, serving a broad spectrum of applications for spatiotemporal control of cells and tissues in 2D and 3D. We hope that Reviewer #3 will agree with our rational argumentation and appreciate the value of addressing distinct groups of target readers.